# Budget-Optimized Crowdworker Allocation

**Sha Lai**                                                                    *lais823@bu.edu*
*Department of Computer Science*
*Boston University*

**Margrit Betke**[*]                                                           *betke@bu.edu*
*Department of Computer Science*
*Boston University*

**Prakash Ishwar**                                                             *pi@bu.edu*
*Department of Electrical and Computer Engineering*
*Boston University*

**Reviewed on OpenReview:** *https://openreview.net/forum?id=hVpAgznRmp*

## Abstract

Due to concerns about human error in crowdsourcing, it is standard practice to collect labels for the same data point from multiple internet workers. We show that the resulting budget can be used more effectively with a *flexible* worker assignment strategy that asks fewer workers to analyze data that are easy to label and more workers to analyze data that requires extra scrutiny. Our main contribution is to show how the worker label aggregation can be formulated using a probabilistic approach, and how the allocations of the number of workers to a task can be computed optimally based on task difficulty alone, without using worker profiles. Our representative target task is identifying entailment between sentences. To illustrate the proposed methodology, we conducted simulation experiments that utilize a machine learning system as a proxy for workers and demonstrate its advantages over a state-of-the-art commercial optimizer.

## 1 Introduction

Machine learning research is advanced by crowdsourcing efforts, for example, to generate training data and evaluate machine learning models (Wortman Vaughan, 2018). When deciding on how many internet workers to employ to annotate data, crowd task organizers must strike a compromise between budget constraints and accuracy expectations. Multiple annotations are typically collected for the same data point, out of concern for the accuracy of human annotation (Kovashka et al., 2014). Building this redundancy into the crowdsourcing experiment, however, increases its cost and cannot guarantee accuracy.

The literature describes techniques for computing optimal trade-offs between accuracy and redundancy in crowdsourcing using a *fixed* number of crowd workers per task (Karger et al., 2013; Tran-Thanh et al., 2013). The fixed assignment is agnostic about the latent difficulty of each task, i.e., it is data independent. In this work, our focus is on a *flexible* and *data-dependent* assignment scheme. Intuitively, tasks can be categorized based on labeling difficulty: "easy" tasks are associated with a high probability of correct labeling, "hard" tasks have a low probability, and those in between require more scrutiny. While assigning extra workers to "hard-to-label" tasks seems like a way to improve correctness, it is often inefficient. Often, for truly hard tasks, even experts may disagree, meaning more worker annotations do not necessarily yield better results and can waste resources. Similarly, easy tasks require minimal input. Thus, fewer internet workers should analyze easy- and hard to-label data, and more workers should analyze the remaining data.

---

[*]This author passed away prior to the publication of this work.

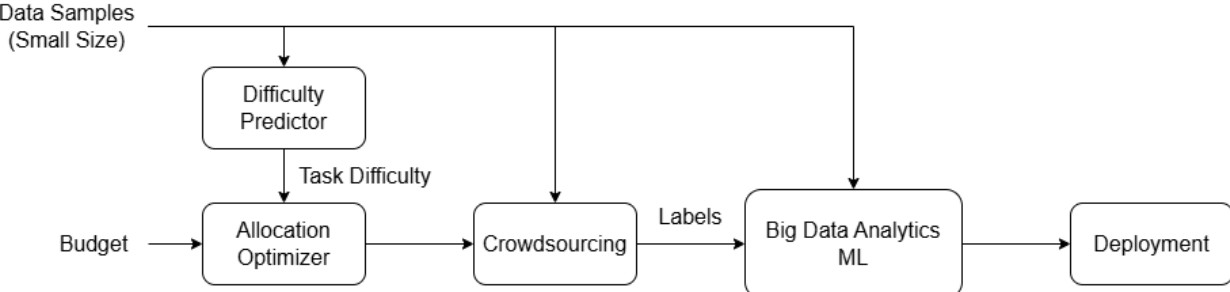

Figure 1: Human-Machine Labeling Framework. First, the task difficulty of each sample is assessed and used to optimize the crowdsourcing procedure. Second, the labels from the crowdworkers are used for analysis and training a ML model, which can then be deployed for large-scale tasks.

Flexible schemes have not been adequately explored in the literature (Sameki et al., 2016a;b; Khetan & Oh, 2016). Handcrafted decision-trees (Sameki et al., 2016a) and random-forest predictors (Sameki et al., 2016b) have been developed to determine how many crowd workers to assign to a labeling task. It was shown that the computed worker allocations can provide large budget savings with small sacrifices in accuracy compared to the traditional fixed allocation scheme. We propose an alternative approach to determine the number of crowd workers per task that directly *optimizes* the allocation over all data. For a given budget, we show that the optimal allocation of the number of crowd workers results in both accuracy improvements and budget savings.

Consider the human-machine labeling framework shown in Figure 1. The system first answers the question: How many crowdworkers should be assigned to label the data so as to achieve the best average labeling accuracy while not exceeding the target budget? The solution to this question provides a budget-optimized crowdsourcing method to accurately label a small dataset. This initial labeled dataset then enables the development of an ML model through a training-validation-test cycle. Finally, the developed model can be deployed for label production where the model is used to classify a large number of data samples. We propose a new crowdsourcing methodology as a tool for ML researchers such as the ones who work on computer vision or natural language processing domains relying on manual image, video, or text annotations via crowdsourcing to train their recognition systems (Nowak & Rüger, 2010; Rashtchian et al., 2010; Russell et al., 2008; Sawant et al., 2011; Su et al., 2012; Vijayanarasimhan & Grauman, 2011; Yan et al., 2010). Following our methodology, the envisioned human-machine system could yield data analysis at a scale even beyond what is possible with large crowdsourcing efforts.

Focusing on the first phase of the human-machine labeling framework shown in Figure 1, we make three contributions in this paper:

- The problem of determining a budget-optimal flexible crowdsourcing strategy is formulated as an optimization problem. The variables in our formulation represent the worker-to-task allocations, the sum of the worker costs is constrained to be upper-limited by a budget, and the objective function measures the average accuracy of aggregated worker labels. We first focus on the setting where the task labels are binary and then address the general scenario.

- For crowdsourcing tasks with binary labels, we propose the budget-optimized crowdworker allocation (BUOCA) algorithm that can efficiently find an optimal solution to the corresponding optimization problem. BUOCA is a greedy algorithm that runs in polynomial time.

- We extend our binary formulation to the multiclass scenario. Our simulation experiments show the advantages of our multiclass BUOCA algorithm compared to a commercial optimization solver in terms of label quality and time-efficiency.

## 2   Related Works

In recent years, the meaning and the focus of crowdsourcing has shifted from the practice of hiring online workers to annotate or analyze data to almost any online information-gathering practice such as Wikipedia article creation (Raković, 2023), online voting (Wijekoon et al., 2022), Internet of Things framework (Sodagari, 2022), and so on. While these practices share a similar process as the crowdsourcing for data annotation, the nature of these tasks create task-specific variables, resulting in very different scenarios when researchers analyze and study their procedures. Therefore, we wish to clarify that this study focuses specifically on the original model of crowdsourcing: the recruitment of online workers for data annotation.

Balancing the demands that accuracy requirements and budget limits place on crowdsourcing experiments has been the focus of research in various communities, including machine learning (Chen et al., 2015; Dai et al., 2010; Gao et al., 2016; Karger et al., 2013; 2014; Kolobov et al., 2013; Manino et al., 2018; Simpson & Roberts, 2014; Tran-Thanh et al., 2013), human computation (Gurari & Grauman, 2017; Li & Liu, 2015; Sameki et al., 2016a), data management (Bansal et al., 2016; Davtyan et al., 2015; Gao et al., 2013; Parameswaran et al., 2012; Wu et al., 2018), and computer vision (Gurari et al., 2016; 2018; Jain & Grauman, 2013). The crowdsourcing mechanisms used in practice, e.g., collecting image labels to train computer vision systems, are typically agnostic to the difficulty of a task, assigning the same fixed number of crowdworkers to each task. Notable exceptions are the recent works by Gurari et al. (2018), Sameki et al. (2016a), and Gurari & Grauman (2017), who proposed flexible worker assignment schemes.

If experience ratings of crowdworkers exist and the difficulty of a task can be discerned, routing easy tasks to novice workers and difficult tasks to expert annotators has also been proposed (Kolobov et al., 2013; Karger et al., 2014). Optimal task routing, however, is an NP-hard problem, and so online schemes for task-to-worker assignments have been proposed (Bragg et al., 2014; Chang et al., 2015; Fan et al., 2015; Rajpal et al., 2015). Recently, the difficulty of a crowdsourcing task has been linked to its ambiguity (Gurari et al., 2018). For some datasets, there may not be "correct" but only subjective labels.

Our work is different from previously-proposed crowdsourcing methodologies with adaptive worker assignments (Chen et al., 2013a; Dai et al., 2010; Simpson & Roberts, 2014; Tran-Thanh et al., 2013; Welinder & Perona, 2010) because these assume that the same workers can be employed with "user profile tracking." The worker-task allocation scheme by Dai et al. (2010) relies on being able to "incrementally estimate [the workers' accuracy] based on their previous work." The algorithm by Tran-Thanh et al. (2013) relies on a majority-voting-efficient "fusion method to estimate the answers to each of the tasks," which also requires user profile tracking. Our methodology does not include user profile tracking because in our experiments using the Amazon Mechanical Turk Internet marketplace, we cannot request the same workers in an incremental scheme to estimate the accuracy of their work. Our work makes use of maximum likelihood estimate under certain conditions (Section 3.2).

Moreover, our work is distinct from prior works that employ a Markov Decision Process (MDP) (Li et al., 2016b; Chen et al., 2013b; Dai et al., 2013). These works model the crowdsourcing procedure using an MDP model which estimates, in real time, the next optimal action given the worker annotations in each time step of a crowdsourcing experiment. Our work, on the other hand, is a "batch method" that focuses on estimating the optimal number of workers required for each task before publishing the crowdsourcing jobs to the workers.

## 3   Proposed Crowdsourcing Methodology

In this section, we describe our budget-optimized crowdworker allocation problem and our proposed method to solve it. Our problem is formulated in terms of a combinatorial optimization program with an objective function that captures labeling accuracy and a budget that depends on crowdworker allocations. We first focus on binary labeling tasks and then discuss the general multiclass case. For binary tasks we establish the optimality of a greedy allocation strategy when the decision fusion accuracy satisfies certain conditions.

### 3.1 Formulation of Crowdsourcing Optimization

The goal of the budget-optimized crowdworker allocation problem is to maximize a measure of overall labeling accuracy while ensuring that a measure of overall worker cost does not exceed a target budget. This can be formulated as an optimization problem where the worker allocations to different data samples are positive integer variables, the overall labeling accuracy across data samples is the optimization objective, and the total worker budget is the constraint.

#### 3.1.1 Variables

In order to formalize our approach, let the data samples be indexed by the positive integers $i$ in the range $\{1, \ldots, I\}$ and let $n_i$ denote the number of crowdworkers allocated to the $i$-th data sample. Let $\boldsymbol{n} = (n_1, n_2, ..., n_I)^\top$ denote the $I$-tuple of crowdworker allocations. We assume that every data sample is allocated at least one crowdworker and not more than $N$ crowdworkers. Thus, for all $i$, we have $n_i \in \{1, \ldots, N\}$, i.e., $\boldsymbol{n} \in \{1, \ldots, N\}^I$.

#### 3.1.2 Crowdsourcing cost and budget

We only consider additive costs in this work. Specifically, the cost function $f_{cost}$ of the crowdsourcing experiment for crowdworker allocation tuple $\boldsymbol{n}$ is defined by

$$f_{\text{cost}}(\boldsymbol{n}) = f_{\text{cost}}(n_1, n_2, ..., n_I) := \sum_{i=1}^{I} c \cdot n_i$$

where $c$ is the unit cost per sample per crowdworker. We denote the total budget by $B$. Thus, $f_{\text{cost}}(\boldsymbol{n}) \leq B$.

#### 3.1.3 Crowdsourcing accuracy

We next define the measure of accuracy that we use. Let $Q(i, n_i)$ denote the probability that the decisions of $n_i$ crowdworkers when combined is correct for the $i$-th sample, *i.e.*, it matches the ground truth of the $i$-th sample. Let $Q$ be a matrix (table) with $I$ rows and $N$ columns with value $Q(i, n_i)$ denoting the value in the $i$-th row and the $n_i$-th column. Then, we can measure the overall accuracy of all tasks given allocation-tuple $\boldsymbol{n} = (n_1, n_2, ..., n_I)^\top$ as the expected correct labeling rate (CLR) averaged across all samples:

$$\text{CLR}(Q, \boldsymbol{n}) := \frac{1}{I} \sum_{i=1}^{I} Q(i, n_i). \tag{1}$$

We describe the computation of $Q(i, n_i)$ in the following sub-sections.

#### 3.1.4 Problem statement

The budget-optimized crowdworker allocation problem is the following combinatorial optimization problem:

$$\boldsymbol{n}_*(Q, B, I) := \arg \max_{\substack{\boldsymbol{n} \in \{1, ... N\}^I \\ f_{\text{cost}}(\boldsymbol{n}) \leq B}} \text{CLR}(Q, \boldsymbol{n}) \tag{2}$$

### 3.2 Decision Fusion Method

The value of $Q(i, n_i)$ depends on the method used to combine decisions of crowdworkers as well as the underlying joint probability distribution of ground truth labels and labels assigned by crowdworkers. A variety of models and methods for data aggregation have been studied in the literature, e.g., the survey by Li et al. (2016a). In particular, some complex models can account for noise, expertise, and bias of annotators, but they have higher computational complexity and require additional information which may

not be available. For example, annotator reliability on Figure Eight is based on previously completed tasks which may not indicate reliability on a new, completely unrelated task.

Therefore, we consider a different approach which is focused on the difficulty of the labeling task rather than worker factors. Specifically, we make the following assumptions which are motivated by the real-world unavailability of fine-grained information about task-difficulties, labeling noise, or the expertise, biases, and correlation of annotators, etc. For example, on real-world crowdsourcing platforms like Amazon Mechanical Turk, worker correlation information is not available for quality optimization.

**Assumption 3.0.1. (Crowdworkers)** *The crowdworkers are statistically indistinguishable and make independent decisions. In particular, the crowdworkers make independent and identically distributed (iid) decisions when annotating each sample.*

**Assumption 3.0.2. (Samples)** *Each sample in the dataset has a single ground truth label and the samples together with their ground truth labels, are sampled in an iid manner according to some underlying distribution.*

**Assumption 3.0.3. (Model)** *We model the difficulty of labeling the $i$-th sample as $p_i$, the probability of the said sample being labeled correctly by a randomly chosen crowdworker. The value of $p_i$ depends on the sample $i$, but we assume that it is the same no matter what the ground truth label of the sample is or which crowdworker labels it.*

A high value of $p_i$ indicates an easy labeling task, whereas a low value of $p_i$ indicates a hard labeling task.

To clarify Assumption 3.0.3, we note that each task is composed of two parts: the content, e.g., text, image, etc., and the label. The assumption means that the probability of correct labeling depends directly on the content and not on the label.

In Assumption 3.0.1, the probability that a decision will be correct depends on the sample. For any given sample, the probability of correctly labeling it will be the same for any crowdworker (statistical indistinguishability of crowdworker decision-making), but the value of that probability will depend on the content of the sample being annotated.

In Assumption 3.0.2, by "are sampled in an iid manner according to some underlying distribution," we mean that all workers receive tasks completely at random (neither the task requester nor the platform will inject any rules in distributing the tasks).

Additionally, we assume, for now, that the sample labels are binary, i.e., the label space is $\mathcal{Y} = \{1, 2\}$. The extension to multi-class cases will be discussed later in Section (6).

In what follows, we will show, for the $i$-th task with $n_i$ worker labels, how to infer the most likely label from worker annotations using the Maximum Likelihood Estimation (MLE), and how to calculate the probability that the inferred label is correct in terms of $p_i$. This probability is precisely $Q(i, n_i)$ under our approach.

### 3.2.1 Maximum Likelihood Estimate of Label from Crowdworker Annotations

Suppose data sample $i$ has been labeled by $n_i$ crowdworkers. Let the number of workers deciding classes 1 and 2 be denoted by $m_i^{(1)}$ and $m_i^{(2)}$ respectively and let $M_i = (m_i^{(1)}, m_i^{(2)})$. Note that $m_i^{(1)} + m_i^{(2)} = n_i$. For convenience we define $\mathcal{M}(n)$ as the set of all non-negative integer tuples that sum to $n$, i.e.,

$$\mathcal{M}(n) := \{(m^{(1)}, m^{(2)}) : m^{(1)}, m^{(2)} \in \{0, 1, \ldots, n\}, m^{(1)} + m^{(2)} = n\}.$$

Then $M_i \in \mathcal{M}(n_i)$. Let $l_i \in \mathcal{Y} = \{1, 2\}$ represent the true label of data sample $i$. Then, the probability of observing $M_i$ given $l_i$ is given by the following:

$$P(M_i | l_i = 1) = \binom{n_i}{m_i^{(1)}} p_i^{m_i^{(1)}} (1 - p_i)^{n_i - m_i^{(1)}}$$

$$P(M_i | l_i = 2) = \binom{n_i}{m_i^{(2)}} p_i^{m_i^{(2)}} (1 - p_i)^{n_i - m_i^{(2)}}$$

or more compactly by:

$$P(M_i|l_i) = \binom{n_i}{m_i^{(l_i)}} p_i^{m_i^{(l_i)}} (1-p_i)^{n_i - m_i^{(l_i)}} \tag{3}$$

The maximum likelihood estimate of the true label for data sample $i$ given $M_i$ is given by

$$\hat{y}_{\text{MLE}}^{(i)}(M_i) = \underset{y \in \{1,2\}}{\arg\max} \; P(M_i|y) \tag{4}$$

$$= \underset{y \in \{1,2\}}{\arg\max} \; \binom{n_i}{m_i^{(y)}} p_i^{m_i^{(y)}} (1-p_i)^{n_i - m_i^{(y)}}$$

$$= \underset{y \in \{1,2\}}{\arg\max} \; p_i^{m_i^{(y)}} (1-p_i)^{n_i - m_i^{(y)}}$$

$$= \underset{y \in \{1,2\}}{\arg\max} \; (m_i^{(y)} \log(p_i) + (n_i - m_i^{(y)}) \log(1-p_i))$$

$$= \underset{y \in \{1,2\}}{\arg\max} \; h(y, M_i, p_i) \tag{5}$$

where the second equality follows from the combinatorial identity $\binom{n_i}{m_i^{(1)}} = \binom{n_i}{n_i - m_i^{(1)}} = \binom{n_i}{m_i^{(2)}}$ and

$$h(y, M_i, p_i) := m_i^{(y)} \log(p_i) + (n_i - m_i^{(y)}) \log(1-p_i). \tag{6}$$

We assume that if there is a tie between $y = 1$ and $y = 2$ in Equation (5), then the tie is broken by choosing a label in $\{1, 2\}$ uniformly at random.

If the binary ground truth labels are equally likely, then the maximum likelihood estimate of the label from $M_i$ is also the Maximum Aposteriori Probability (MAP) estimate of the label based in $M_i$ which minimizes the probability of estimation error. For simplicity and since the true label probabilities will not be available, we use the maximum likelihood Estimate and make the following assumption:

**Assumption 3.0.4.** *(Ground truth label) All values of the ground truth label are equally likely.*

### 3.3 Computation of Q: MLE Accuracy

Let $E_i$ denote the event that $\hat{y}_{\text{MLE}}^{(i)}(M_i)$ matches the ground truth $l_i$, i.e., $\hat{y}_{\text{MLE}}^{(i)}(M_i) = l_i$. Then, $Q(i, n_i) = P(E_i)$. When $p_i = 0.5$, the probability of $E_i$ is trivially 0.5. When $p_i \neq 0.5$, we can compute the $P(E_i)$ as follows:

$$P(E_i) = \sum_{l_i} \sum_{M_i \in \mathcal{M}(n_i)} P(E_i|M_i, l_i) \cdot P(M_i|l_i) \cdot P(l_i). \tag{7}$$

In Equation (7), we only need to consider the choices of $M_i$ such that $P(E_i|M_i, l_i) > 0$, i.e., choices of $M_i$ for which $\hat{y}_{\text{MLE}}^{(i)}(M_i) = l_i$ or equivalently by (5) only $M_i$ such that the ground truth label $l_i$ has the maximum likelihood:

$$\forall y_i \neq l_i, \; h(y_i, M_i, p_i) \leq h(l_i, M_i, p_i). \tag{8}$$

Anticipating that there could be $M_i$ for which the likelihood of another label may be tied with that of $l_i$ for the maximum, we define

$$J_{n_i, l_i, p_i, d} := \left\{ M_i \in \mathcal{M}(n_i) \; \middle| \; \begin{array}{l} \forall y_i \neq l_i, h(y_i, M_i, p_i) \leq h(l_i, M_i, p_i), \\ \text{with equality only for } (d-1) \text{ values of } y_i \neq l_i \end{array} \right\} \tag{9}$$

as the set of $M_i$ for which there are $d$ labels whose likelihoods are tied for the maximum with the ground truth label $l_i$ and where $d$ can range from 1 through $|\mathcal{Y}| = 2$. For all $M_i \in J_{n_i, l_i, p_i, d}$, we have

$$P(E_i|M_i, l_i) = \frac{1}{d}$$

We can now rewrite Equation (7) using $J_{n_i,l_i,p_i,d}$ as follows:

$$P(E_i) = \sum_{l_i=1}^{2} \sum_{d=1}^{2} \sum_{M_i \in J_{n_i,l_i,p_i,d}} \frac{1}{d} \cdot P(M_i|l_i) \cdot P(l_i) \tag{10}$$

By Assumption 3.0.4, $P(l_i)$ follows a uniform distribution, i.e., $P(l_i) = \frac{1}{2}$. Moreover, the value of $p_i$ is the same no matter what the ground truth label of the sample (Assumption 3.0.3) and the crowdworkers make iid decisions (Assumption 3.0.1). It therefore follows that $P(E_i|l_i)$ is the same for all values of $l_i$. Therefore, Equation (10) can be simplified by replacing $l_i$ with an arbitrary label $l_0$ and removing the outer summation:

$$P(E_i) = \sum_{d=1}^{2} \sum_{M_i \in J_{n_i,l_0,p_i,d}} \frac{1}{d} \cdot P(M_i|l_0) \tag{11}$$

Finally, we can use (3) to express the quantity $P(M_i|l_0)$. By doing so, we conclude the calculation of $P(E_i)$, which, by definition, is exactly the value of $Q(i, n_i)$:

$$Q(i,n_i) = P(E_i) = \begin{cases} 0.5, & \text{if } p_i = 0.5 \\ \sum_{d=1}^{2} \sum_{M_i \in J_{n_i,l_0,p_i,d}} \frac{1}{d} \cdot \binom{n_i}{m_i^{(l_0)}} \cdot p_i^{m_i^{(l_0)}} \cdot (1-p_i)^{n_i-m_i^{(l_0)}}, & \text{otherwise} \end{cases} \tag{12}$$

## 4 Greedy Algorithm for Crowdworker Allocation and its Optimality

The optimization problem defined in Section 3.1 is in general NP-hard. For the case when the task labels are binary, we propose a greedy algorithm that finds an optimal solution in polynomial time.

### 4.1 Budget-Optimized Crowdworker Allocation Algorithm

The algorithm, which we name as "BUdget-Optimized Crowdworker Allocation" (BUOCA) algorithm, takes input parameters $Q$, $B$ and $c$. The first parameter $Q$ is the table storing $Q(i, n_i)$ values, while the second parameter $B$ and the third one $c$ are the budget and the unit cost respectively. The algorithm returns $\boldsymbol{n}$, a list of optimal worker allocations for all tasks, as outputs. The pseudo code is described in Algorithm 1.

---

**Algorithm 1** BUOCA Algorithm (Q, B, c)

---

1: Initialize array $\boldsymbol{n}$ where each entry $n_i$ stores the worker allocation for task $i$
2: Set all worker allocations (all entries of $\boldsymbol{n}$) to be 1
3: Initialize the starting cost $\beta = cI$
4: **while** $\beta < B$ **do**
5:     Find $\lambda = \arg\max_i (Q(i, n_i + 2) - Q(i, n_i))$
6:     Set $\beta = \beta + 2c$
7:     **if** $\beta \leq B$ **then**
8:         Set $n_\lambda = n_\lambda + 2$
9:     **end if**
10: **end while**
11: Return $\boldsymbol{n}$

---

Intuitively, the algorithm starts with the allocation of one worker per sample. This corresponds to a total allocation of $I$ workers with a cost of $cI$. Then, the algorithm increases the total allocation by two workers step-by-step till the cost reaches the budget $B$. By the definition of CLR in Equation (1), when two additional workers are allocated to task $i$, the CLR value changes by the amount $\frac{1}{I}(Q(i, n_i + 2) - Q(i, n_i))$. Therefore, at each step $m$, the algorithm chooses the task sample $\lambda$ for additional allocation that results in the largest

increase in CLR as follows:

$$\lambda = \arg\max_i \left( \frac{1}{I}(Q(i, n_i + 2) - Q(i, n_i)) \right) = \arg\max_i (Q(i, n_i + 2) - Q(i, n_i)) \tag{13}$$

The algorithm finds a globally optimal allocation by greedily accumulating locally optimal ones. In the next sub-section, we will show that the resulting worker allocation is indeed the globally optimal allocation.

## 4.2 Global Optimality of BUOCA

In this sub-section, we will prove the global optimality of BUOCA in a sequence of theorems. We first relate our MLE approach to infer label from worker annotations to voting by simple majority, and then show how a simple majority vote leads to the global optimality of BUOCA.

### 4.2.1 Relation between MLE and Simple Majority Voting

We first show that if $p_i$, the probability of task $i$ being correctly labeled by a random worker, is greater than half, then MLE is the same as majority voting.

**Theorem 4.1.** *If $p_i > 0.5$, then inferring the true label by the MLE approach from $n_i$ worker annotations is the same as voting by simple majority.*

*Proof.* Suppose the task sample $i$ is labeled by $n_i$ workers and the worker label class counts are denoted as the tuple $M_i = (m_i^1, m_i^2)$. The maximum likelihood estimation of the true label of the $i$-th sample is given by Equation (5). Let $y_i' := \hat{y}_{\text{MLE}}^{(i)}(M_i)$ for convenience. Then we must have the following property:

$$\text{for all } y_i \neq y_i', \; m_i^{(y_i')} \log(p_i) + (n_i - m_i^{(y_i')}) \log(1 - p_i) \geq m_i^{(y_i)} \log(p_i) + (n_i - m_i^{(y_i)}) \log(1 - p_i)$$

The inequality can be simplified as

$$(m_i^{(y_i')} - m_i^{(y_i)})(\log p_i - \log(1 - p_i)) \geq 0$$

If $p_i > 0.5$, then $(\log p_i - \log(1 - p_i)) > 0$ and hence $(m_i^{(y_i')} - m_i^{(y_i)}) \geq 0$. In other words, $m_i^{(y_i')}$ must be the majority vote in $M_i = (m_i^{(y_i')}, m_i^{(y_i)})$. $\qquad\square$

In a similar manner one can show that if $p_i < 0.5$, then inferring the true label using the MLE approach from $n_i$ worker annotations is the same as simple minority voting.

### 4.2.2 Relation between $p_i$ and $Q(i, n_i)$

As shown in Equation (12), $Q(i, n_i)$ for $p_i \neq 0.5$ is in fact, a function of both $p_i$ and $n_i$. We denote this function by $q(p_i, n_i)$:

$$Q(i, n_i) = q(p_i, n_i) = \sum_{d=1}^{2} \sum_{M_i \in J_{n_i, l_0, p_i, d}} \frac{1}{d} \cdot \binom{n_i}{m_i^{(l_0)}} \cdot p_i^{m_i^{(l_0)}} \cdot (1 - p_i)^{n_i - m_i^{(l_0)}} \tag{14}$$

We next show that $q(p_i, n_i)$ is in fact, symmetric with respect to $p_i$.

**Theorem 4.2.** *Let $q(p_i, n_i)$ be a function as defined in Equation (14), then $q(p_i, n_i) = q(1 - p_i, n_i)$.*

*Proof.* When $p_i = 0.5$ then $1 - p_i = p_i = 0.5$ and trivially we have $q(p_i, n_i) = q(1 - p_i, n_i)$. Now consider the case $p_i \neq 0.5$. We recall the definitions of $h(y, M_i, p_i)$ from Equation (6) and $J_{n_i, l_0, p_i, d}$ from Equation (9):

$$h(y, M_i, p_i) := m_i^{(y)} \log(p_i) + (n_i - m_i^{(y)}) \log(1 - p_i)$$

and

$$J_{n_i, l_i, p_i, d} := \left\{ M_i \in \mathcal{M}(n_i) \,\middle|\, \begin{array}{l} \forall y_i \neq l_i, h(y_i, M_i, p_i) \leq h(l_i, M_i, p_i), \\ \text{with equality only for } (d-1) \text{ values of } y_i \neq l_i \end{array} \right\}.$$

Next observe that $h(1, M_i, p_i) = h(2, M_i, 1 - p_i)$. This implies that $(m_i^{(1)}, m_i^{(2)}) \in J_{n_i, l_i, p_i, d} \Leftrightarrow (m_i^{(2)}, m_i^{(1)}) \in J_{n_i, l_i, 1-p_i, d}$. Therefore, there is a bijection between $J_{n_i, l_i, p_i, d}$ and $J_{n_i, l_i, 1-p_i, d}$ such that for any element $M_i = (m_i^{(1)}, m_i^{(2)}) \in J_{n_i, l_i, 1-p_i, d}$, the corresponding element in $J_{n_i, l_i, p_i, d}$ is $M_i' = (m_i^{(2)}, m_i^{(1)})$. Using this together with the complementary property of binomial distribution, we have

$$
\begin{aligned}
q(1 - p_i, n_i) &= \sum_{d=1}^{2} \sum_{M_i \in J_{n_i, l_0, 1-p_i, d}} \binom{n_i}{m_i^{(l_0)}} \cdot (1 - p_i)^{m_i^{(l_0)}} \cdot p_i^{n_i - m_i^{(l_0)}} \\
&= \sum_{d=1}^{2} \sum_{M_i \in J_{n_i, l_0, 1-p_i, d}} \binom{n_i}{n_i - m_i^{(l_0)}} \cdot p_i^{n_i - m_i^{(l_0)}} \cdot (1 - p_i)^{m_i^{(l_0)}} \\
&= \sum_{d=1}^{2} \sum_{M_i \in J_{n_i, l_0, p_i, d}} \binom{n_i}{m_i^{(l_0)}} \cdot p_i^{m_i^{(l_0)}} \cdot (1 - p_i)^{n_i - m_i^{(l_0)}} \\
&= q(p_i, n_i)
\end{aligned}
\tag{15}
$$

Therefore, we conclude that $q(p_i, n_i) = q(1 - p_i, n_i)$. $\qquad\square$

From Theorem 4.1 and Theorem 4.2, we can assume, without loss of generality, that $p_i > 0.5$ when working with Equation (14). Then, $J_{n_i, l_0, p_i, d}$ can be written as

$$
J_{n_i, l_0, p_i, d} = \begin{cases} \left\{ (m_i^{(1)}, m_i^{(2)}) \,\middle|\, \begin{array}{l} m_i^{(1)}, m_i^{(2)} \in \mathbb{Z}^*; \ m_i^{(1)} + m_i^{(2)} = n_i; \\ \text{for } y_i \neq l_0, \ m_i^{(y_i)} < m_i^{(l_0)} \end{array} \right\}, & \text{if } d = 1 \\[3ex] \{(\frac{n_i}{2}, \frac{n_i}{2})\}, & \text{if } d = 2 \text{ and } n_i \text{ is even} \\[2ex] \varnothing, & \text{if } d = 2 \text{ and } n_i \text{ is odd} \end{cases}
$$

Based on this observation, Equation (14) reduces to

$$
Q(i, n_i) = q(p_i, n_i) = \begin{cases} \displaystyle\sum_{m_i^{l_0} = \frac{n_i+1}{2}}^{n_i} \binom{n_i}{m_i^{l_0}} p_i^{m_i^{l_0}} (1 - p_i)^{n_i - m_i^{l_0}}, & \text{if } n_i \text{ is odd} \\[3ex] \displaystyle\sum_{m_i^{l_0} = \frac{n_i}{2}+1}^{n_i} \binom{n_i}{m_i^{l_0}} p_i^{m_i^{l_0}} (1 - p_i)^{n_i - m_i^{l_0}} + \frac{1}{2} \binom{n_i}{\frac{n_i}{2}} p_i^{\frac{n_i}{2}} (1 - p_i)^{\frac{n_i}{2}}, & \text{otherwise} \end{cases}
\tag{16}
$$

### 4.2.3 Allocation Incremental Size

Before proceeding to prove the global optimality of BUOCA, we would like to use the proven theorems to explain why BUOCA adds two workers for the selected task at each step. The initial number of workers allocated for each task is one, therefore, intuitively, adding two workers at each subsequent step can avoid ties. Nonetheless, we now show that adding only one worker at each step when the current worker number is odd will not change the CLR value. To do so, we first prove the following theorem.

**Theorem 4.3.** *Let* $n \in \mathbb{Z}^+$, $p \in [0, 1]$, *and*

$$
q(p, n) = \begin{cases} \displaystyle\sum_{m = \frac{n+1}{2}}^{n} \binom{n}{m} p^m (1 - p)^{n-m}, & \text{if } n \text{ is odd} \\[3ex] \displaystyle\sum_{m = \frac{n}{2}+1}^{n} \binom{n}{m} p^m (1 - p)^{n-m} + \frac{1}{2} \binom{n}{\frac{n}{2}} p^{\frac{n}{2}} (1 - p)^{\frac{n}{2}}, & \text{otherwise} \end{cases}
$$

*If* $n$ *is odd, then* $q(p, n) = q(p, n + 1)$.

*Proof.* Suppose $n$ is odd, let $a = \frac{1}{2}(n+1)$ and observe that $n - (a-1) = a$. Then, by the recurrence property of binomial coefficients and algebra, we have

$$q(p, n+1) = \frac{1}{2}\binom{n+1}{a}p^a(1-p)^a + \sum_{m=a+1}^{n+1}\binom{n+1}{m}p^m(1-p)^{n+1-m}$$

$$= \frac{1}{2}\left\{\binom{n}{a} + \binom{n}{a-1}\right\}p^a(1-p)^a + \sum_{m=a+1}^{n+1}\left\{\binom{n}{m} + \binom{n}{m-1}\right\}p^m(1-p)^{n+1-m}$$

$$= \binom{n}{a}p^a(1-p)^a + \sum_{m=a+1}^{n+1}\binom{n}{m}p^m(1-p)^{n+1-m} + \sum_{m=a+1}^{n+1}\binom{n}{m-1}p^m(1-p)^{n+1-m}$$

$$= \binom{n}{a}p^a(1-p)^a + (1-p)\sum_{m=a+1}^{n}\binom{n}{m}p^m(1-p)^{n-m} + \sum_{m=a}^{n}\binom{n}{m}p^{m+1}(1-p)^{n-m}$$

$$= \binom{n}{a}p^a(1-p)^a + \sum_{m=a+1}^{n}\binom{n}{m}p^m(1-p)^{n-m}$$

$$\quad - \sum_{m=a+1}^{n}\binom{n}{m}p^{m+1}(1-p)^{n-m} + \sum_{m=a}^{n}\binom{n}{m}p^{m+1}(1-p)^{n-m}$$

$$= \binom{n}{a}p^a(1-p)^a + \binom{n}{a}p^{a+1}(1-p)^{n-a} + \sum_{m=a+1}^{n}\binom{n}{m}p^m(1-p)^{n-m}$$

$$= p\binom{n}{a}p^a(1-p)^{n-a} + (1-p)\binom{n}{a}p^a(1-p)^{n-a} + \sum_{m=a+1}^{n}\binom{n}{m}p^m(1-p)^{n-m}$$

$$= \binom{n}{a}p^a(1-p)^{n-a} + \sum_{m=a+1}^{n}\binom{n}{m}p^m(1-p)^{n-m}$$

$$= \sum_{m=a}^{n}\binom{n}{m}p^m(1-p)^{n-m}$$

$$= q(p, n)$$

$\square$

We now formally state the theorem regarding the incremental size.

**Theorem 4.4.** *Let $Q(i, n_i)$ be defined in Equation (12). For any odd positive integer $n_i$, $Q(i, n_i) = Q(i, n_i + 1)$*

*Proof.* First of all, if $p_i = 0.5$, then $Q(i, n_i) = Q(i, n_i + 1) = 0.5$ is trivially true for all positive integer $n_i$. Now, consider the case when $p_i \neq 0.5$.

By Theorem 4.1 and Theorem 4.2, we can assume, without loss of generality, that $p_i > 0.5$. Then, $Q(i, n_i)$ reduces to the form described in Equation (16). Finally, by Theorem 4.3, it follows that $Q(i, n_i) = Q(i, n_i + 1)$ for any odd positive integer $n_i$. $\square$

### 4.2.4 Concavity and Monotonicity Properties of $q(p, n)$

We next prove the following theorem which establishes the concavity and monotonicity of $q(p, n)$. The theorem not only serves as a stepping stone for the global optimality of BUOCA, but also suggests that, in practice, the while loop of the algorithm can stop early when no positive change of CLR can be made when choosing the next task by Equation (13).

**Theorem 4.5.** *If $n$ is an odd positive integer and $p \in [0, 1]$, then*

$$q(p, n) = \sum_{m=\frac{n+1}{2}}^{n} \binom{n}{m} p^m (1-p)^{n-m}$$

*is strictly increasing and concave with respect to the integer variable $n$ for all $p > 0.5$.*

*Proof.* We shall first prove the monotonicity property of $q(p, n)$.

Since $n$ is an odd positive integer, let $n = (2t - 1)$, where $t$ is a positive integer. Let $X[1], X[2], \ldots$, be independent and identically distributed Bernoulli($p$) random variables. Then $P(X[1] = 1) = p = 1 - P(X[1] = 0)$. Let $S[t] := X[1] + \ldots + X[2t - 1]$ denote the total number of successes in $n = (2t - 1)$ Bernoulli($p$) trials. Finally, let $a[t] := P(S[t] \geq t)$. Then, $q(p, n) = q(p, 2t - 1) = a[t]$.

Since $S[t + 1] = S[t] + X[2t] + X[2t + 1]$, we can express $q(p, n + 2) = a[t + 1] = P(S[t + 1] \geq t + 1)$ as follows:

$$\begin{aligned}
a[t+1] &= P(S[t] + 0 \geq t + 1) \cdot P(X[2t] = 0, X[2t+1] = 0) \\
&\quad + P(S[t] + 1 \geq t + 1) \cdot P(X[2t] = 0, X[2t+1] = 1) \\
&\quad + P(S[t] + 1 \geq t + 1) \cdot P(X[2t] = 1, X[2t+1] = 0) \\
&\quad + P(S[t] + 2 \geq t + 1) \cdot P(X[2t] = 1, X[2t+1] = 1) \\
&= P(S[t] \geq t + 1) \cdot (1-p)^2 + P(S[t] \geq t) \cdot (2p(1-p)) + P(S[t] \geq t - 1) \cdot p^2
\end{aligned}$$

Now, $P(S[t] \geq t) = a[t]$ and $P(S[t] \geq t+1) = a[t] - P(S[t] = t)$. Also, $P(S[t] \geq t-1) = a[t] + P(S[t] = t-1)$. Thus,

$$a[t+1] = a[t] - (1-p)^2 \cdot P(S[t] = t) + p^2 \cdot P(S[t] = t - 1).$$

Since $P(S[t] = s) = \binom{n}{s} p^s (1-p)^{n-s}$ and $n = (2t - 1)$, we have $\binom{n}{t} = \binom{n}{n-t} = \binom{n}{t-1}$ and therefore $P(S[t] = t - 1) = P(S[t] = t) \cdot (1 - p)/p$. Using this together with $a[t + 1] = q(n + 2)$, and $a[t] = q(n)$ in the above equation we obtain:

$$q(p, n + 2) = q(p, n) + P(S[t] = t) \cdot (1 - p)(2p - 1)$$

The last term is positive if $p > 0.5$, negative if $p < 0.5$, and zero if $p = 0.5$. This proves that $q(p, n)$ is strictly increasing if $p > 0.5$, strictly decreasing if $p < 0.5$, and a constant (equal to $q(0.5, 1) = 0.5$) if $p = 0.5$.

We shall now establish the concavity property of $q(p, n)$. From the proof of monotonicity of $q(p, n)$ we have

$$q(p, n + 2) - q(p, n) = P(S[t] = t) \cdot (1 - p)(2p - 1).$$

Then the ratio

$$\begin{aligned}
\frac{q(p, n + 4) - q(p, n + 2)}{q(p, n + 2) - q(p, n)} &= \frac{P(S[t + 1] = t + 1)}{P(S[t] = t)} = \frac{\binom{2t+1}{t+1} p^{t+1}(1-p)^{2t+1-(t+1)}}{\binom{2t-1}{t} p^t (1-p)^{2t-1-t}} \\
&= 2p(1 - p) \cdot \frac{2t + 1}{t + 1}.
\end{aligned}$$

For all positive integers $t$, $2(2t + 1)/(t + 1) < 4$ and for all $p \neq 0.5$ we have $p(1 - p) < 1/4$. It follows that if $p \neq 0.5$, the ratio is strictly less than one. Thus, $q(p, n)$ is strictly concave for all $p \neq 0.5$. $\square$

### 4.2.5 Proof of Global Optimality of BUOCA

The final theorem states the global optimality of BUOCA:

**Theorem 4.6.** *If for all $i$, $Q(i, n_i)$ is either (1) non-increasing or (2) increasing and concave with respect to $n_i$, then the greedy algorithm BUOCA returns globally optimal allocations and CLR values for all budgets from $cI$ to $cNI$, where $c$ is the unit cost, $I$ is the total number of tasks, and $N$ is the maximum workers allowed for each task.*

*Proof.* First we note that the monotonicity and concavity properties in the statement of the theorem only need to hold with respect to *odd* positive integers, as BUOCA only considers these values for $n_i$.

For all data samples $i$ for which $Q(i, n_i)$ is non-increasing, the optimal allocation is clearly equal to 1. The BUOCA algorithm will return this value for these samples since their initial allocation is 1 and they never get incremented because $Q(i, n_i + 2) - Q(i, n_i)$ is never positive. Thus the evolution of the BUOCA algorithm is unaffected by the presence of samples for which $Q(i, n_i)$ is non-increasing.

Therefore, for the purpose of this proof we assume, without loss of generality, that the increasing and concavity conditions in the statement of the theorem are satisfied by *all* data samples. Formally,

$$\forall i \in \{1, \ldots, I\}, \ Q(i, n_i) \text{ is increasing in } n_i. \tag{17}$$

and

$$\forall i \in \{1, \ldots, I\} \text{ and all odd positive integers } n_i,$$
$$Q(i, n_i + 4) - Q(i, n_i + 2) \leq Q(i, n_i + 2) - Q(i, n_i). \tag{18}$$

Let us denote the iteration index (step) of BUOCA with variable $m$, the allocations at step $m$ as $\boldsymbol{n}[m]$, and the cost at step $m$ as $\beta[m]$. Since the total allocation is incremented at each step, proving the global optimality of BUOCA for all budgets from $cI$ to $cNI$ is equivalent to proving that when $B = cNI$, $\mathrm{CLR}(Q, \boldsymbol{n}[m])$ is optimal for all $m$.

We have shown in Section 4.1 that the CLR value is increased step-by-step in BUOCA and the improvement is calculated as

$$\mathrm{CLR}(Q, \boldsymbol{n}[m+1]) - \mathrm{CLR}(Q, \boldsymbol{n}[m]) = \frac{1}{I}(Q(\lambda, n_\lambda[m] + 2) - Q(\lambda, n_\lambda[m]))$$

where $I$ is the total number of tasks and $\lambda$ is selected according to Equation (13).

Since the $Q(i, n_i)$'s are increasing in $n_i$ for all $i$, by construction, $\mathrm{CLR}(Q, \boldsymbol{n}[m])$ is increasing in $m$ as well. The increasing trend only stops when the cost $f_{\mathrm{cost}}(n[m]) = \sum_{i=1}^{I} c \cdot n_i[m]$ reaches the budget $B$, and we denote that final time step with $m_f$.

For convenience, we let $\boldsymbol{\delta}_i$ denote the tuple of length $I$ whose $i$-th component is 1 and all other components are 0. The proof is by induction on $m$.

**Base case (step 1):** For $m = 1$, since there is only one feasible list of allocations namely: $n_1[1] = \ldots = n_I[1] = 1$, it is trivially globally optimum for the cost $\beta[1] = cJ$.

If $m_f = 1$ we are done. If $m_f > 1$, then for $m < m_f$ we have the following induction hypothesis.

**Induction hypothesis (step $m$):** Let $\boldsymbol{n}[m]$ returned by BUOCA be globally optimum. Then, for all $\boldsymbol{n}'[m]$ that are feasible under the same cost,

$$\mathrm{CLR}(Q, \boldsymbol{n}[m]) \geq \mathrm{CLR}(Q, \boldsymbol{n}'[m]).$$

**Inductive step $m+1$:** Let $\boldsymbol{n}'[m+1]$ be any feasible list of allocations for the next higher cost of $\beta[m+1] = \beta[m] + 2c$ corresponding to two additional crowdworkers and let $\boldsymbol{n}[m+1]$ be the corresponding allocation list returned by BUOCA. We will demonstrate that $\mathrm{CLR}(Q, \boldsymbol{n}[m+1]) \geq \mathrm{CLR}(Q, \boldsymbol{n}'[m+1])$ and thereby prove the result.

Since $m < m_f$, for $\lambda$ given by Eq. (13), we have

$$\boldsymbol{n}[m+1] = \boldsymbol{n}[m] + 2\boldsymbol{\delta}_\lambda \tag{19}$$

and for all $i \in \{1, \dots, I\}$,

$$\mathrm{CLR}(Q, \boldsymbol{n}[m] + 2\boldsymbol{\delta}_\lambda) \geq \mathrm{CLR}(Q, \boldsymbol{n}[m] + 2\boldsymbol{\delta}_i). \tag{20}$$

It is sufficient to consider only $\boldsymbol{n}'[m+1]$ that have the same cost as $\boldsymbol{n}[m+1]$ which, since $m < m_f$, equals $\beta[m+1] = \beta[m] + 2c$. This is because any lower cost feasible allocation list will have a CLR that is less than or equal to the CLR of $\boldsymbol{n}[m]$ which, by the induction hypothesis, is globally optimum for its cost. Since $m < m_f$, the CLR of $\boldsymbol{n}[m]$ is in turn strictly dominated by the CLR of $\boldsymbol{n}[m+1]$.

Hence we can focus on $\boldsymbol{n}'[m+1]$ such that $\sum_{i=1}^{I} n_i'[m+1] = \sum_{i=1}^{I} n_i[m+1]$. If $\boldsymbol{n}'[m+1] \neq \boldsymbol{n}[m+1]$ then there is at least one sample $j \in \{1, \dots, I\}$ for which $n_j'[m+1] > n_j[m+1]$ or equivalently, since allocations can only be odd positive integers,

$$n_j'[m+1] \geq n_j[m+1] + 2 \geq n_j[m] + 2 \tag{21}$$

where the last inequality is because in step $m$, allocations increase by 2 for sample $\lambda$ and do not increase for all other samples. This leads us to the following series of inequalities

$$
\begin{aligned}
\mathrm{CLR}(Q, \boldsymbol{n}[m+1]) \stackrel{(a)}{=} \; & \mathrm{CLR}(\boldsymbol{n}[m] + 2\boldsymbol{\delta}_\lambda) \\
\stackrel{(b)}{\geq} \; & \mathrm{CLR}(\boldsymbol{n}[m] + 2\boldsymbol{\delta}_j) \\
\stackrel{(c)}{=} \; & \mathrm{CLR}(\boldsymbol{n}[m]) + \frac{1}{I}(Q(j, n_j[m] + 2) - Q(j, n_j[m])) \\
\stackrel{(d)}{\geq} \; & \mathrm{CLR}(\boldsymbol{n}'[m+1] - 2\boldsymbol{\delta}_j) + \frac{1}{I}(Q(j, n_j[m] + 2) - Q(j, n_j[m])) \\
\stackrel{(e)}{\geq} \; & \mathrm{CLR}(\boldsymbol{n}'[m+1] - 2\boldsymbol{\delta}_j) + \frac{1}{I}(Q(j, n_j'[m+1]) - Q(j, n_j'[m+1] - 2)) \\
\stackrel{(f)}{=} \; & \mathrm{CLR}(\boldsymbol{n}'[m+1])
\end{aligned}
$$

where $(a)$ follows from Equation (19), $(b)$ from Equation (20), $(c)$ and $(f)$ follow from the definition of CLR in Equation (1), $(d)$ follows from the global optimality of $\boldsymbol{n}[m]$ for the cost $\beta[m]$ and the fact that the cost of $\boldsymbol{n}'[m+1]$ equals $\beta[m] + 2c$, and finally, inequality $(e)$ follows from Equation (21) and the concavity condition of Equation (18). To explain inequality $(e)$ in more detail: $n_j[m], n_j[m] + 2$ are two consecutive odd integers and so are $n_j'[m+1] - 2, n_j'[m+1]$. Equation (21) shows that the pair $n_j'[m+1] - 2, n_j'[m+1]$ is not to the left of (i.e., is greater than or equal to) the pair $n_j[m], n_j[m] + 2$. The concavity condition of Equation (18) shows that the $Q$ function increment over all consecutive odd integers can only decrease or remain constant as we consider larger and larger pairs of odd consecutive integers. This implies inequality $(e)$. $\qquad \square$

## 5 Simulations to Demonstrate Effectiveness of BUOCA

We conducted simulated experiments to demonstrate the effectiveness of the BUOCA algorithm. Recall that the inputs to BUOCA are the budget $B$ and the $Q$ table containing the probabilities $\{p_i\}$ of correct combined worker decisions for all allocations of workers and all samples. Given $Q$, BUOCA provably yields optimal crowdworker allocations for any budget $B$.

While users of BUOCA may have a good idea about the maximum budget $B$ they can expend, they may not know how difficult some tasks may be for a given pool of crowdworkers. Consequently, the $\{p_i\}$ values that make up the $Q$ table are not directly available in practice and would need to be estimated from training data using a suitable ML model. The main goal of this section is to explore the impact that inaccuracies in

the estimation of the $\{p_i\}$ values have on the CLR of allocations found by the BUOCA algorithm. Figure 2 shows the simulation workflow in detail. The simulation workflow takes a dataset, including sample contents such as images, text, etc., and true labels (one for each sample). An ML model, such as a neural network, is then trained on the labeled dataset and then used to assign **soft labels**, i.e., the class probability values of the final softmax layer (or equivalent values if the ML model is not a neural network) to any given input sample.

The workflow then splits into two branches, each producing a CLR score. According to the top branch, for each sample $i$, the soft label entry corresponding to the correct (ground truth) discrete label of that sample is chosen as the probability of labeling (i.e., the $p_i$ value). These probability values are then used to construct the $Q$ table, denoted by $Q_T$, which is used to determine the ideal optimal allocation $A_T$. Lastly, a CLR score is computed based on $Q_T$ and $A_T$. In the bottom branch of the workflow, for each sample $i$, the soft labels are used to create simulated crowdworker labels: $N$ crowdworker labels are generated randomly according to the soft label probability distribution for that sample and then the fraction of correct ones together with the ground truth label of the sample is used to construct the table $Q_S$ which is then used to determine the optimal allocations $A_S$. Lastly, a CLR score is computed with $Q_T$ and $A_S$.

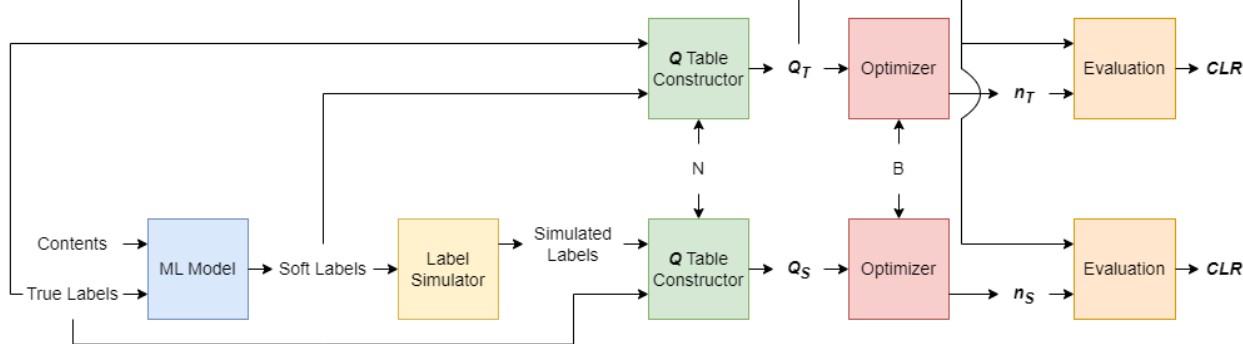

Figure 2: Simulation Workflow

Once the workflow is completed, we can evaluate the performance of BUOCA when run on potentially inaccurate input obtained from the ML model by comparing the two CLR scores (defined in Equation (1)): $\text{CLR}(Q_T, A_T)$ for the top branch indicating the performance based on using the true $Q$ table, and $\text{CLR}(Q_T, A_S)$ for the bottom branch indicating the performance based on using a $Q$ table estimated from simulated crowdworker labels. Note that for the bottom branch, the $Q$ table used for evaluation is the one from the top branch since the top branch corresponds to the true $Q$ table in these experiments.

The whole workflow is repeated for a range of budget values so that we can observe a performance trend (except we only need to train the model once). For each tested budget value, We repeatedly simulate the bottom branch 100 times to estimate confidence bounds for the CLR evaluation.

**Experiments Using Simulation Workflow on Textual Entailment Task**

We applied the workflow described previously to a Recognizing Textual Entailment (RTE) dataset. The original dataset (namely, *RTE1*) was released by Dagan et al. (2005). However, we used a subset of the original dataset which was released by Snow et al. (2008) and has been used by previous crowdsourcing works. There are 800 task samples in the RTE dataset each of which contains a pair of sentences, namely text and hypothesis, and a binary ground truth label annotated by experts indicating whether or not the truth of the text leads to the truth of the hypothesis. Exactly half of the 800 samples are labeled negative while the other half are labeled positive.

To generate simulated worker labels we used a pre-trained DeBERTa model released by Sileo (2023). Instead of re-training the model on the dataset, we performed fine-tuning. The fine-tuning accuracy scores for

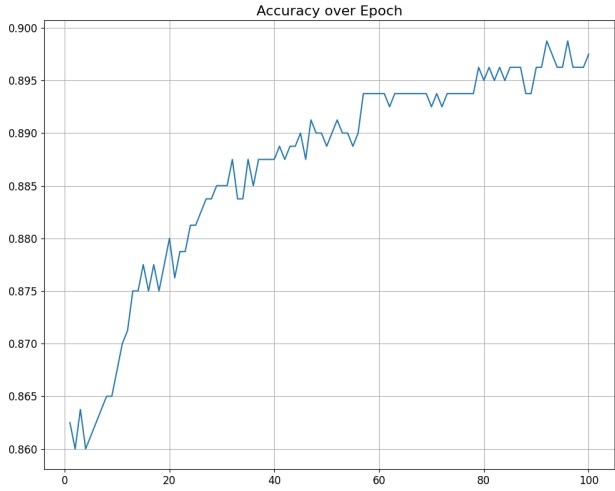

Figure 3: Fine-tuning accuracy

different numbers of epochs are recorded in Figure 3. The accuracy scores vary between 0.86 and 0.9, with higher accuracy for higher epoch numbers.

We varied two impactful parameters during the experiments: the epoch number for which the DeBERTa model was fine-tuned, and $N$, the maximum number of simulated annotations of each task. In theory, both parameters can affect the CLR values: the epoch number affects the model's prediction power and hence the correct labeling probability for each task, and the maximum number of simulated annotations directly impacts the $Q$ table.

**Discussion of Results**

Our experiments suggest that both the number of fine-tuning epochs and the number of simulated crowd-worker annotations show positive correlation with the CLR score. Plots of CLR scores with respect to budget for different numbers of epochs and different maximum numbers of simulated crowdworkers are shown in Figure 4.

The plots in the same row of Figure 4 share the same fine-tuning epoch number, while the plots in the same column have the same maximum number $N$ of simulated per-task annotations. In each plot, the blue curve corresponds to the CLR values obtained with the model's soft labels (top branch in Figure 2), the orange curve corresponds to the mean CLR values obtained with the simulated annotations (bottom branch in Figure 2), and the green and the red curves show the standard deviation of the mean CLR values. For all plots, the budget ranges from single worker assignment per task to maximum worker assignment. The "uniform-allocation budget point" is the end budget point in each curve and corresponds to assigning $N$ crowdworkers to each task.

As these plots show, increasing the number of fine-tuning epochs or the number of simulated crowdworkers not only leads to greater CLR values but also narrows the gaps between the ideal values (blue curve) and the mean values based on the ML prediction (orange curve). The DeBERTa model instance we chose was so well trained that fine-tuning for only three epochs already yields a high performance score and increasing the number of epochs quickly leads to diminishing improvements. For example, with the maximum number of workers set to 3, the highest ideal CLR value achieved is 0.95 using a model fine-tuned for merely three epochs (first row and first column of Figure 4). After fine-tuning for two more epochs (second row and first column of Figure 4), the CLR increases by only 0.01. On the other hand, increasing the maximum number of simulated crowdworkers per task seems to create slightly more noticeable changes to the CLR score. Take the bottom row in Figure 4 for example. When the maximum number of workers per task is 3, the highest ideal CLR value is about 0.971, which is also the highest CLR value in the same column of the

figure. Nonetheless, increasing the maximum number of workers to 9 pushes the CLR up to about 0.984 and increasing $N$ to 20 nudges the the highest CLR value to nearly 0.99.

Additionally, in all of the plots in Figure 4 there is a budget value after which the CLR improvement with increasing budget becomes negligible. For example, with the maximum number of workers set to 9 and a model fine-tuned for 5 epochs (second row and second column of Figure 4), the ideal CLR value begins exhibiting diminishing improvements around a budget of 2000 and the curve stays nearly flat for budgets greater than 3500. Should the user desire to stop the execution of BUOCA at some point before the cost reaches the total budget due to diminishing improvements, they could easily modify the algorithm by introducing a threshold for the marginal increase in CLR value.

In general, the phenomenon of diminishing improvements of CLR with increasing budgets is a consequence of the (provably) concave shape of the ideal CLR versus budget curve. Any decision to increase the current budget will depend on the steepness of the concave-shaped tradeoff curves and will differ from application-to-application. In some cases, the rise may be steep and in others, not so much. As a rough practical recommendation, it would be highly beneficial to increase the budget in the regimes where the slope is high.

## 6 Multi-Class Extension

Thus far, we focused on binary task labels. We now show how to extend the binary label formulation to the multi-class scenario.

### 6.1 MLE and Inference Accuracy

The extension starts with a new quantity $p_{i,x|l_i}$, the probability of task $i$ having ground truth label $l_i$ being labeled as class $x$ by a random worker. We maintain Assumption 3.0.1 and Assumption 3.0.2.

Next, we again use MLE to infer the true label for given $n_i$ worker labels. Let $C$ be the number of label classes and let $M_i = (m_i^{(1)}, m_i^{(2)}, ..., m_i^{(C)})$ be the tuple of class counts of the $n_i$ worker labels. We infer the true label for a given $M_i$ similarly to Equation (4).

The calculation of $P(M_i|l_i = y)$ needs to be adjusted to the multi-class scenario as follows:

$$P(M_i|l_i = y) = \binom{n_i}{m_i^{(1)} m_i^{(2)} ... m_i^{(C)}} \prod_{x=1}^{C} p_{i,x|y}^{m_i^{(x)}} \tag{22}$$

This leads to the new expression for $\hat{y}_{\text{MLE}}^{(i)}(M_i)$:

$$\begin{aligned}
\hat{y}_{\text{MLE}}^{(i)}(M_i) &= \arg\max_{y} P(M_i|l_i = y) \\
&= \arg\max_{y} \binom{n_i}{m_i^{(1)} m_i^{(2)} ... m_i^{(C)}} \prod_{x=1}^{C} p_{i,x|y}^{m_i^{(x)}} \\
&= \arg\max_{y} \prod_{x=1}^{C} p_{i,x|y}^{m_i^{(x)}} \\
&= \arg\max_{y} \sum_{x=1}^{C} m_i^{(x)} \log(p_{i,x|y})
\end{aligned} \tag{23}$$

The calculation of $P(E_i)$, the probability that $\hat{y}_{\text{MLE}}^{(i)}(M_i)$ matches the ground truth $l_i$, is very similar to the binary scenario. Nevertheless, the set $J_{n_i,l_i,d}$ needs to be modified as follows

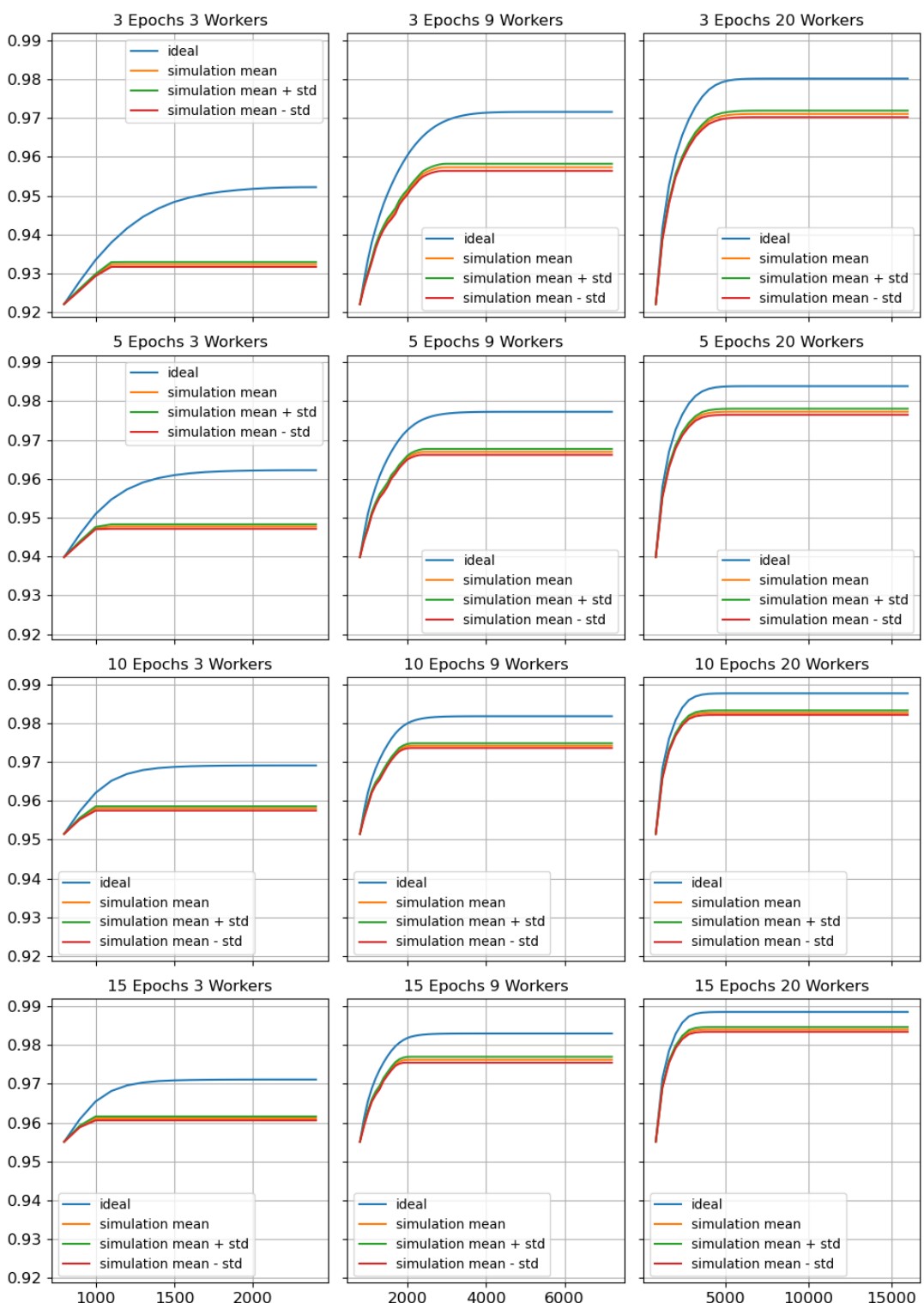

Figure 4: CLR-vs-budget plots for binary-label tasks.

$$J_{n_i, l_i, p_{i, \cdot | l_i}, d} = \left\{ M_i \in \mathcal{M}(n_i) \;\middle|\; \begin{array}{l} \forall y \neq l_i, h(l_i, M_i, p_{i, \cdot | l_i}) \geq h(y, M_i, p_{i, \cdot | y}), \\ \text{with equality only for } (d-1) \text{ values of } y \neq l_i \end{array} \right\}$$

where

$$\mathcal{M}(n) := \{(m^{(1)}, m^{(2)}, \ldots, m^{(C)}) : m^{(1)}, m^{(2)}, \ldots, m^{(C)} \in \{0, 1, \ldots, n\}, m^{(1)} + m^{(2)} + \ldots + m^{(C)} = n\}.$$

and

$$h(y, M_i, p_{i,\cdot|y}) = \sum_{x=1}^{C} m_i^{(x)} \log(p_{i,x|y}).$$

Lastly, the new form of $Q(i, n_i) = P(E_i)$ is

$$Q(i, n_i) = P(E_i) = \begin{cases} \dfrac{1}{C}, & \text{if } p_{i,\cdot|l_i} = \dfrac{1}{C} \text{ for any } l_i \\ \displaystyle\sum_{d=1}^{C} \sum_{M_i \in J_{n_i, l_0, p_{i,\cdot|l_0}, d}} \dfrac{1}{d} \binom{n_i}{M_i} \prod_{x=1}^{C} p_{i,x|l_0}^{m_i^{(x)}}, & \text{otherwise} \end{cases} \tag{24}$$

This completes the calculation of $Q(i, n_i)$ in the multi-class label scenario.

## 6.2 Simplification

The calculation shown in Equation (24) requires the knowledge of all $p_{i,x|y}$ values. This may not be practical in practice. However, one can simplify the calculation by making the following assumption:

**Assumption 6.0.1.** *For each task $i$,*

$$p_{i,x|y} = \begin{cases} p_i & \text{if } x = y \\ \frac{1-p_i}{C-1} & \text{otherwise.} \end{cases}$$

Under this assumption, we can again use the quantity $p_i$, the probability of task $i$ being correctly labeled by a random worker, to express $Q(i, n_i)$:

$$Q(i, n_i) = P(E_i) = \begin{cases} \dfrac{1}{C}, & \text{if } p_i = \dfrac{1}{C} \\ \displaystyle\sum_{d=1}^{C} \sum_{M_i \in J_{n_i, l_0, p_{i,\cdot|l_0}, d}} \dfrac{1}{d} \binom{n_i}{M_i} p_i^{m_i^{(l_0)}} \left(\dfrac{1-p_i}{C-1}\right)^{n_i - m_i^{(l_0)}}, & \text{otherwise} \end{cases} \tag{25}$$

## 6.3 Adapted BUOCA Algorithm

The original BUOCA algorithm shown in Algorithm 1 is not suitable for the multiclass scenario as it is designed for binary tasks. For example, the result that the worker allocation increment can be restricted to two without loss of optimality (Theorem 4.4 ) would not hold, in general, for non-binary tasks. We therefore propose a modified greedy algorithm, Multiclass BUOCA, described in Algorithm 2. In Multiclass BUOCA, the worker increment size is not held fixed in advance to a specific value (like one or two).

As demonstrated in Section 4.2, the global optimality of BUOCA algorithm for binary tasks hinged on the $Q(i, n_i)$ function being either non-increasing or increasing and concave with respect to $n_i$ for all $i$. The multiclass $Q(i, n_i)$, unfortunately, does not enjoy these properties in general. Therefore, unlike BUOCA, Multiclass BUOCA is not guaranteed to yield a globally optimal solution; only a locally optimal one (being a greedy algorithm).

## 6.4 Simulation Experiments for Multiclass BUOCA

We conducted simulation experiments following the same workflow described in Section 5, with the ML Model, the Label Simulator and the Optimizer changed for the multi-class scenario. Since Multiclass BUOCA

---

**Algorithm 2** Multiclass BUOCA Algorithm (Q, B, c)

---

1: Initialize array $\boldsymbol{n}$ where each entry $n_i$ stores the worker allocation for task $i$
2: Set all worker allocations (all entries of $\boldsymbol{n}$) to be 1
3: Initialize the starting cost $\beta = cI$
4: Denote $I$ as the number of tasks (i.e., the number of rows in $Q$)
5: Denote $N$ as the maximum number of workers of each task (i.e., the number of columns in $Q$)
6: **while** $\beta < B$ **do**
7:    Initialize array $\boldsymbol{s}$ of length $I$ where each entry $s_i = 0$
8:    Initialize array $\boldsymbol{d}$ of length $I$ where each entry $d_i = 0$
9:    **for** each sample $i$ **do**
10:       Find the smallest integer $k \in (0, N - n_i]$ $s.t.$ $Q(i, n_i + k) - Q(i, n_i) > 0$
11:       Set $s_i = k$ and $d_i = Q(i, n_i + k) - Q(i, n_i)$
12:    **end for**
13:    Find $\lambda = \arg\max_i(\boldsymbol{d})$
14:    Set $\beta = \beta + c \cdot s_\lambda$
15:    **if** $\beta \leq B$ **then**
16:       Set $n_\lambda = n_\lambda + s_\lambda$
17:    **end if**
18: **end while**
19: Return $\boldsymbol{n}$

---

is not guaranteed to provide the globally optimal solution, we reformulated the optimization problem as an equivalent Integer Programming (IP) problem and compared the CLR versus budget curves and associated runtimes of Multiclass BUOCA and the state-of-the-art commercial IP solver Gurobi (Gurobi Optimization, LLC, 2024).

### Experiments

For our multiclass experiments we used a combination of two RTE datasets: *RTE2* (Haim et al., 2006) and *RTE3* (Giampiccolo et al., 2007). These two datasets contain the same five entailment labels, one of which is "no relation." Since the samples labeled as "no relation" occupy about half of the dataset, creating a rather unbalanced scenario, we kept only the samples with one of the remaining four labels for simulation. This led to a dataset of 810 samples and the label ratios $205 : 187 : 206 : 212$ which is close to uniform. The ML model used was again the pre-trained DeBERTa model released by Sileo (2023), but with a classifier layer for four classes.

Moreover, in order to use the IP solver, we constructed an IP problem equivalent to Equation (2) as follows.

$$
\begin{aligned}
\max \quad & \sum_{i=1}^{I} \sum_{j=1}^{N} x_{ij} \cdot Q(i,j) \\
\text{s.t.} \quad & \sum_{j=1}^{N} x_{ij} = 1 && i = 1, 2, ..., I \\
& \sum_{i=1}^{I} \sum_{j=1}^{N} x_{ij} \cdot j \leq B \\
& \forall i \forall j, \ x_{ij} = 0, 1
\end{aligned}
\tag{26}
$$

In this IP problem, each binary variable $x_{ij}$ is an indicator for an entry in the $Q$ table. The first constraint ensures that only one entry in each row of the $Q$ table is selected, while the second constraint corresponds to the budget limit.

To compare the efficiency of the two optimization methods, we timed their execution processes when they were computing the "target" scores. Since there are thousands of budget points in these experiments, we selected 40 points equispaced between the lowest and the highest budget points and timed the execution between all consecutive budget points. When given a budget point $B$, Multiclass BUOCA being a greedy algorithm can compute the (possibly suboptimal) allocations for *all* budget points up to $B$. In contrast, the Gurobi IP solver can compute an optimal allocation only for the given budget point $B$. We therefore compare runtimes of Gurobi and Multiclass BUOCA in terms of both the **marginal** execution time between

two consecutive budget points of interest as well as the **accumulative** execution time from the starting budget point.

### Discussion of Results

Our experiments show that, while Multiclass BUOCA was slightly outperformed by the Gurobi solver in terms of CLR when $N$ is small, it was much more efficient than the Gurobi solver, especially when there are more budget points of interest.

We compare the accuracy of Multiclass BUOCA and the commercial solver in CRL-versus-budget plots for different maximum numbers of simulated crowdworkers in Figure 5. The three plots in the first row of Figure 5 show the results from Multiclass BUOCA while the ones in the second row show the results from the Gurobi solver. All six plots are obtained using the same DeBERTa model fine-tuned for 100 epochs. There is a noticeable difference between the plots in the first column corresponding to a maximum of three workers per task. However, the difference becomes insignificant in the other two columns. This observation suggests that when the maximum number of workers per task is not too limited, the suboptimal results from Multiclass BUOCA only trail behind the optimal ones from the commercial solver by a very small margin. Moreover, as in the binary case, we observe diminishing CLR improvements with increasing budgets, although unlike in the binary case we have not mathematically proved that the ideal CLR versus budget curve will always be concave. As in the binary case, a user could incorporate a threshold on the CLR improvement into the algorithm to stop early.

We compare the execution times of the Multiclass BUOCA and Gurobi solver for different maximum numbers of simulated crowdworkers in Figure 6. From these plots, we observe that it takes the Multiclass BUOCA and Gurobi solver similar amounts of time to find the optimal allocation if only a few budget points are given. The runtime of Gurobi solver, however, will skyrocket if many budget points are being considered.

## 7 Conclusions

We modeled the optimal worker allocation problem in crowdsourcing as an optimization problem that aims to maximize aggregated label quality while under a budget constraint. We proposed a probabilistic approach based on the Maximum Likelihood principle to model the decision fusion from workers during crowdsourcing. We showed that if the task labels are binary, then our proposed approach is equivalent to voting by simple majority.

We contributed a new algorithm, BUOCA, which can be used to conduct pilot crowdsourcing studies in order to compute the average correct labeling rate of crowd workers for a given budget and dataset. We proved the global optimality of BUOCA for binary task labels. Further, we showed that Multiclass BUOCA, an adaptive version of BUOCA for multiclass task labels, achieves near-optimal allocations while costing only a fraction of the time it takes for a commercial optimization solver to find the optimal solutions. We would, however, like to note that the near-optimality of Multiclass BUOCA is purely an empirical demonstration. Theoretically quantifying the suboptimality of BUOCA for the general multiclass setting by developing approximation guarantees or data-dependent bounds is an important direction for future research. Lastly, the pilot study results can be used to estimate, for a given budget, the expected accuracy of the results in subsequent larger crowdsourcing studies, where collecting expert labels is prohibitively expensive.

Other than the suboptimal property of Multiclass BUOCA, a limitation of our approach, is the challenge to estimate task difficulty accurately. This limitation applies to both binary and multi-class task settings. Since the task difficulty is the cornerstone of our formulation, a reliable mechanism to accurately estimate task difficulty without incurring additional cost is essential to bring our proposed approach into practice. The design of such mechanisms is an important direction for future work.

We also modeled costs as linear and identical across items. Some deployments face heterogeneous item costs, batch overheads, and latency constraints, which are not addressed in this work.

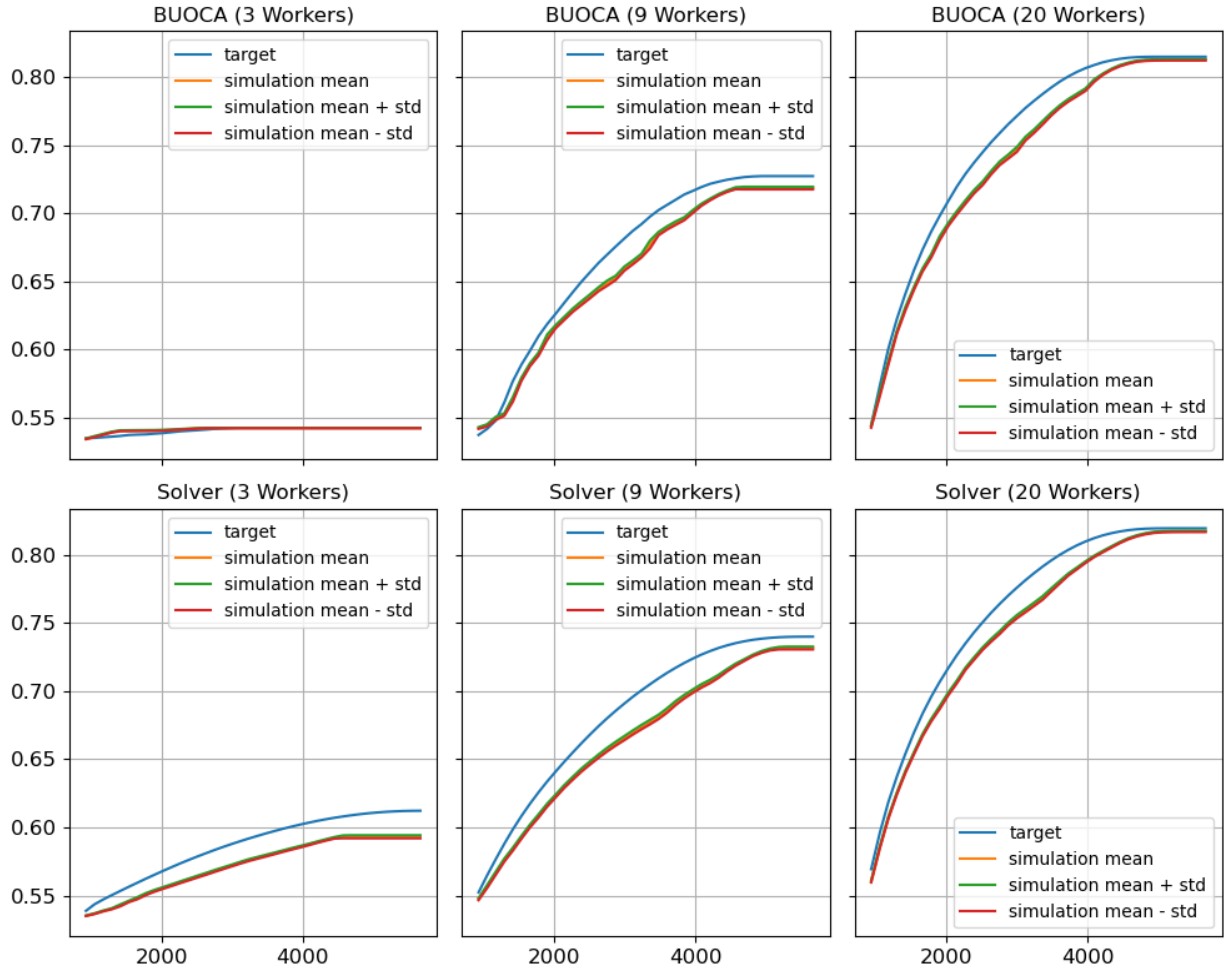

Figure 5: CLR-vs-budget plots for multiclass tasks.

# 8 Broader Impact Concerns

The proposed budgeting strategy can change how platforms value human annotators. Aggressive cost minimization may lower per task pay or reduce available work, which raises fairness concerns and may conflict with platform policies or local labor expectations. The reliance on a single difficulty score for each item can encode model bias and may direct more resources to items that match majority patterns while neglecting minority or sensitive cases. If used in safety critical domains such as medical triage or content moderation, an error driven by miscalibrated difficulty estimates could lead to harmful outcomes for affected users. The work does not discuss privacy risks when difficulty is estimated with models trained on user generated data. The simulation setting does not evaluate impacts on worker well being such as fatigue, rejection rates, or unfair quality control.

# 9 Acknowledgments

We dedicate this work to the memory of our beloved late co-author, Prof. Margrit Betke, who saw this work through completion, but sadly passed away before its publication. This work stands as a tribute to her enduring legacy in the field of Computer Science and Artificial Intelligence. We thank all the reviewers and the Action Editor for their valuable feedback. The Broader Impact Concerns section is entirely the contribution of one of the reviewers.

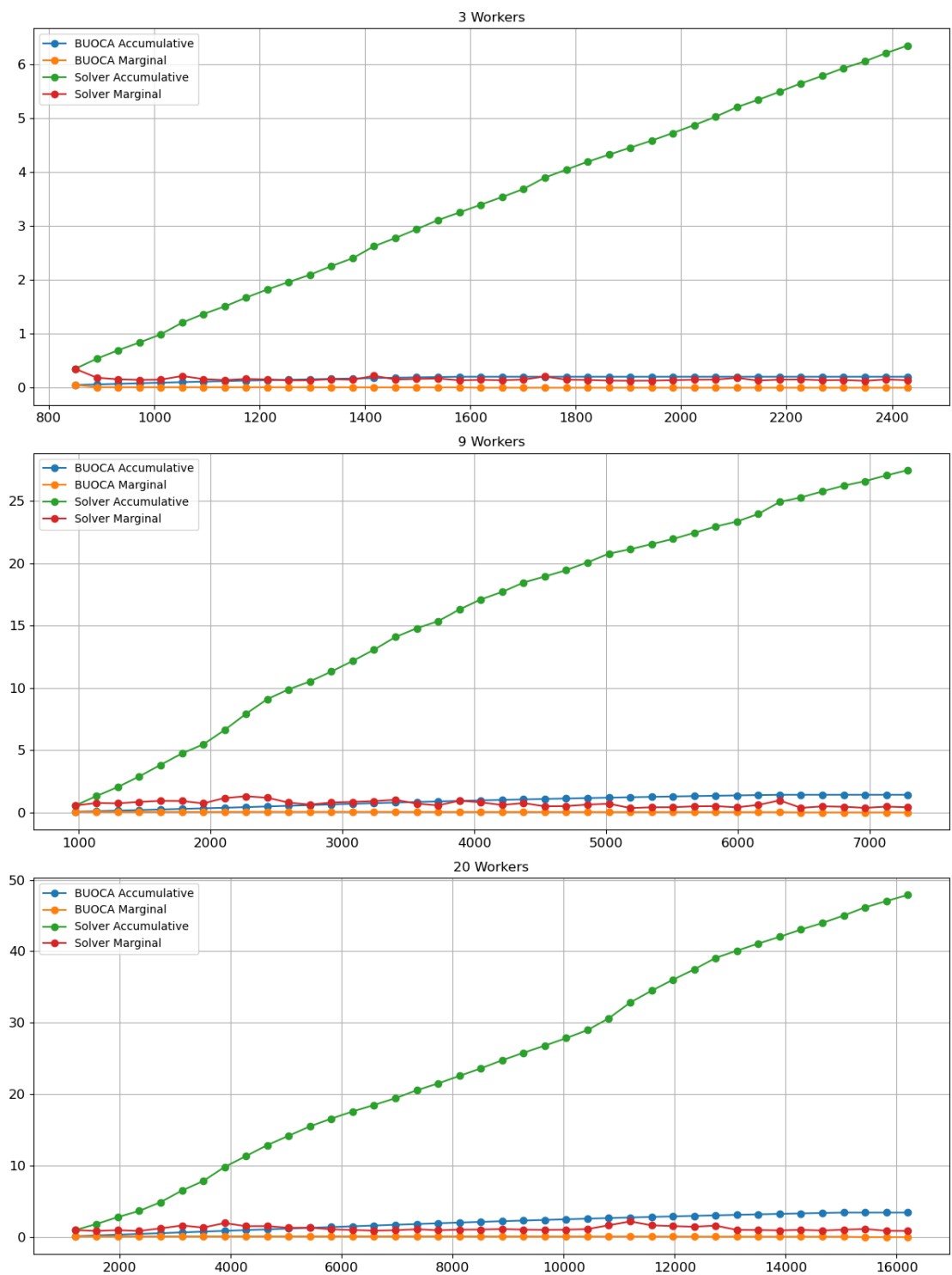

Figure 6: Runtime (in milliseconds) vs. budget plots for multiclass tasks.

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

# A  Appendix

## A.1  Notations

To improve readability, we summarize important notations in Table 1 for quick reference.

Table 1: Notations

| Notation | Meaning |
|---|---|
| $i$ | sample/task index |
| $I$ | number of samples/tasks |
| $C$ | number of classes |
| $n_i$ | number of workers for task $i$ |
| $N$ | maximum number of workers for a task |
| $\boldsymbol{n}$ | $(n_1, n_2, ..., n_I)^\top$, that is, worker allocation list |
| $M_i$ | $(m_i^{(1)}, m_i^{(2)}, ..., m_i^{(C)})$, the class counts of a certain list of worker labels for task $i$ |
| $p_i$ | probability of correct labeling task $i$ by a random worker |
| $p_{yx}$ | probability of labeling a task with label $x$ given the ground truth label $y$ |
| $g_y$ | probability of labeling a task correctly given the ground truth label $y$ |
| $Q(i, n_i)$ | probability of correct labeling task $i$ by the aggregation of $n_i$ random workers |

