# OpenReview forum: "Budget-Optimized Crowdworker Allocation"
_TMLR — Accepted by TMLR_

### Review · Reviewer_LCGK · 2025-10-18

**Summary Of Contributions:**

This paper proposes a method to optimize on worker assignment for crowd sourcing data labeling. The authors formulated this problem into a combinator optimization problem with an objective of labeling accuracy with a budget constraints. The authors firstly model the MLE accuracy with multiple workers and propose a greedy algorithm for crowd worker allocations. The authors further shows the global optimality of BUOCA and demonstrate the method can be applied to multi-class problem as well. The authors conducted experiments with simulated workers.

**Audience:**

Yes

**Audience Explanation:**

Some fields such as data optimization and uncertainty quantification can be potentially benefited by the research.

**Broader Impact Concerns:**

There is no immediate border impact on the proposed method as a general machine learning method.

**Claims And Evidence:**

No

**Claims Explanation:**

The simulated experiment is over simplified with three workers and limited amount of data points. The performance improvements are also marginal. The reviewer cannot directly confirm the validity of the proposed method as there is no direct comparison with different conditions or previous methods/ baselines.
It’s also less intuitive to the reader that how the accuracy of MLE method is computed without knowing the labels from crowdsourcing labels. There have been many studies on uncertainties, active learning and active sampling that tries to quantify the uncertainty in model predictions or optimize data budget for machine learning without knowing the ground truth label. It’s unclear if they share similarities with the proposed method to help understand the validity of the proposed method.

**Requested Changes:**

The writing can be improved as it’s hard to follow the keybindings on definition of MLE accuracy etc. The introduction also did not cover how the method is presented to solve the problems. The proposed method would be more sound if it’s validated by large-scale real-world experiments.

---

> ### Author Response · Authors · 2025-11-12
> **Round 1 Response to Reviewer LCGK**
>
> **Re. “The simulated experiment is over simplified with three workers and limited amount of data points.”**
>
> As explained in the second para of Sec. 5, the main objective of our experiments is to demonstrate the functioning of the BUOCA algorithm and explore the impact that inaccuracies in the estimation of the correct-labeling probabilities have on the CLR of allocations found by the BUOCA algorithm.
>
> The simulation experiments start with the three workers, and then the number of workers is increased to demonstrate the impact of the annotator size on the label quality. Figure 4 compares tradeoff curves for three workers, nine workers, and twenty workers (not just three workers).
>
> The two datasets used in the binary and multi-class experiments contain about 800 samples and ten crowdworker labels per sample. Could the reviewer clarify what they mean by “limited” ?  Does this adjective apply to the number of samples or the number of workers? While the sample size is relatively small when compared to modern machine learning datasets, the labels are real (not simulated) and include expert ground truth labels in addition to annotations from real crowdworkers which are essential for our experiments.
>
> Could the reviewer perhaps suggest larger and publicly available datasets germane to this work?
>
> **Re. “The performance improvements are also marginal … there is no direct comparison with different conditions or previous methods/ baselines.”**
>
> We are unsure what the reviewer meant by “improvements”? Could the reviewer kindly clarify? As explained  in the section on Related Works, our work is markedly different from previously-proposed crowdsourcing methodologies with adaptive worker assignments. We are not aware of any baselines to compare to. Our work focuses on analyzing the properties of BUOCA, a greedy worker allocation algorithm, under a tractable theoretical model whose assumptions are motivated by the unavailability of fine-grained information about task-difficulties, labeling noise, or the expertise, biases, and reliability of annotators, etc. We are not proposing a new crowdsourcing framework.
>
> **Re. “It’s also less intuitive to the reader that how the accuracy of MLE method is computed without knowing the labels from crowdsourcing labels.”**
>
> We assume that by “without knowing the labels from crowdsourcing labels”, the reviewer is referring to “ground truth labels”. As mentioned in the section titled “Simulations to Demonstrate Effectiveness of BUOCA,” indeed the ground truth labels are not available in practice and it may require a machine learning model to (empirically) estimate the probabilities of correct labeling (which can translate to MLE accuracy). The main objective of our experiments is to demonstrate the functioning of the BUOCA algorithm and explore the impact that inaccuracies in the estimation of the correct-labeling probabilities have on the CLR of allocations found by the BUOCA algorithm.
>
> **Re. “There have been many studies on uncertainties, active learning … validity of the proposed method”.**
>
> We would very much appreciate some pointers to specific references related to the methods described by the reviewer. Methods like active learning require real-time calculation during a crowdsourcing experiment. However, as we point out in the last paragraph of the section of Related Works, our work is a “batch method" that focuses on estimating the optimal number of workers required for each task before publishing the crowdsourcing jobs to the workers.
>
> **Re. “The writing can be improved … etc.”**
>
> We are unable to understand what specifically was unclear in our definitions and would appreciate a more detailed elaboration of the difficulties in our exposition.
>
> **Re. “The introduction also did not cover … solve the problems”**
>
> In the introduction section, we clearly state that “the problem of determining a budget-optimal flexible crowdsourcing strategy is formulated as an optimization problem” and we “propose the budget-optimized crowdworker allocation (BUOCA) algorithm” to solve the problem. Could the reviewer kindly clarify what “problems” they are referring to?
>
> **Re. “The proposed method would be more sound if it’s validated by large-scale real-world experiments.”**
>
> Our paper is theory-focused. In order to test our approach on any dataset, the dataset needs to at least contain a sufficiently large number of worker labels and ground truths. We are not aware of any significantly larger, publicly available dataset with this property. One could potentially create a new large dataset by carefully designing and conducting new crowdsourcing experiments. This, however, is a highly nontrivial effort and outside the scope of the present work.

---

> > ### Comment · Reviewer_LCGK · 2025-11-30
> >
> > Thank you for the explanation. I confused the curves in Figure 4. and 5 as comparisons against previous methods. With regard to writing, the reviewer got confused about the introduction while reading the paper because it does not provide an overview about how the problem is approached except for a board "BUOCA" term. It might be different style of writing but the reviewer believes a good introduction would make easy to understand the whole picture of the story. Besides, there have been research on batch active learning such as BatchBALD and "Batch Active Learning at a scale", the reviewer understands the differences in the problem setup. However, it's still concerning that there is no baseline or applicable real-world data for a machine learning literature with direct real-world application scenarios.

---

### Review · Reviewer_39Dj · 2025-11-02

**Summary Of Contributions:**

**Summary**

The paper studies budget aware crowd labeling and frames worker allocation as an optimization problem that maximizes the expected correct labeling rate under a hard budget. Item difficulty is represented by a per item probability that a random worker answers correctly. From these probabilities the authors build a table $Q(i,n)$ giving the expected accuracy after aggregating n annotations for item $i$. They show how $Q(i,n)$ can be derived from maximum likelihood aggregation and its equivalence to majority voting in the binary case. Based on this structure they propose BUOCA, a greedy algorithm that starts with one worker per item and repeatedly assigns two additional workers to the item with the largest marginal gain in expected accuracy until the budget is exhausted. For multiclass labels they design Multiclass BUOCA that increases the number of workers by the smallest positive integer that improves Q for some item. Multiclass BUOCA is not guaranteed to be globally optimal but is efficient. Experiments follow a simulation workflow on textual entailment datasets. A fine-tuned DeBERTa model serves as a proxy for single worker accuracy. The results show that increasing model quality and the maximum workers per item improves the accuracy versus budget curves. Multiclass BUOCA is close to the Gurobi optimum when per item worker caps are not tight and is much faster in runtime across many budget points.

**Strengths**

- The problem is formulated with clarity by maximizing expected accuracy under an explicit budget constraint, which keeps the objective and constraints easy to interpret.

- A precise link between maximum likelihood aggregation and majority voting in the binary case grounds the definition of Q(i,n) and supports later analysis.

- BUOCA in the binary setting is simple to implement yet enjoys a global optimality guarantee under stated conditions, which gives the method strong theoretical footing.

- Runtime efficiency is highlighted through an allocation path that naturally delivers solutions for many intermediate budgets, which is valuable for planning studies.

- The multiclass extension is designed with a clear greedy rule and is compared against an exact integer program, providing a credible upper bound reference for accuracy.

- Experimental design follows a transparent simulation pipeline that varies model quality and per item worker caps, making trends on accuracy versus budget straightforward to read.

- Reproducibility is aided by a self contained setup for building the Q table from difficulty proxies, which lowers the barrier for future comparisons and ablations.


**Weakness**

1. This paper frames the goal as maximizing expected accuracy under a fixed budget. The analysis surveys multiple budget points, yet it does not present a clear Pareto frontier with marginal gains or guidance on how to choose a budget in practice.

2. In this study item difficulty is represented by a per item correctness probability and workers are treated as homogeneous and independent. The modeling choice is neat, but there is no robustness check for correlated annotators, systematic bias, or learning effects.

3. The proposed binary case theory is complete and well argued. External validity remains uncertain because the assumptions that enable optimality are strong and may not hold in many platforms.

4. For the multiclass setting the manuscript introduces a greedy allocator that is efficient in runtime. The absence of approximation guarantees or data dependent bounds leaves the potential suboptimality unquantified beyond solver based comparisons.

5. The experimental setup relies on textual entailment simulations with a classifier used as a proxy for worker accuracy. Evidence from real platforms and from other task families is missing, which limits confidence in generalization.

6. Difficulty is estimated with a fine tuned language model and better proxies yield better curves. Calibration quality, estimation error, and distribution shift are not measured, so the sensitivity of the method to proxy quality is unclear.

7. Costs are modeled as linear and identical across items. Many deployments face heterogeneous item costs, batch overheads, and latency constraints, which are not addressed in the current analysis.

8. Figures report averages and spreads across budgets. Confidence intervals at representative points and simple significance tests are not provided, and diminishing returns are not quantified for decision making.

**Audience:**

Yes

**Audience Explanation:**

Parts of the TMLR audience that work on crowdsourcing, label aggregation, and resource allocation would be interested. The paper gives a clear objective and an optimal greedy method for binary labels. It also offers an efficient multi-class variant that is close to an exact solver in simulations. The results help readers who plan annotation budgets and need accuracy versus budget guidance. The methods and curves can serve as baselines for future work on budget aware labeling.

**Broader Impact Concerns:**

The proposed budgeting strategy can change how platforms value human annotators. Aggressive cost minimization may lower per task pay or reduce available work, which raises fairness concerns and may conflict with platform policies or local labor expectations. The reliance on a single difficulty score for each item can encode model bias and may direct more resources to items that match majority patterns while neglecting minority or sensitive cases. If used in safety critical domains such as medical triage or content moderation, an error driven by miscalibrated difficulty estimates could lead to harmful outcomes for affected users. The work does not discuss privacy risks when difficulty is estimated with models trained on user generated data, nor does it cover transparency to workers about how allocations are decided. The simulation setting does not evaluate impacts on worker well being such as fatigue, rejection rates, or unfair quality control.

**Claims And Evidence:**

Yes

**Claims Explanation:**

The binary setting includes a clear derivation of Q(i,n) from maximum likelihood aggregation and a proof that BUOCA is optimal under stated monotonicity and concavity conditions. The multiclass setting is evaluated against an exact integer program that serves as a credible upper bound, and the greedy method is shown to be close in accuracy while much faster. The simulation pipeline is transparent and varies both model quality and the cap on workers, which makes the accuracy versus budget trends convincing and easy to read.

That said, the evidence is narrower than ideal. Assumptions about homogeneous and independent workers are strong and robustness is not tested. The evaluation is simulation based on textual entailment and does not include a real platform study or additional task families. Multi-class results lack approximation guarantees or data dependent bounds. These gaps limit generality but do not undermine the main claims as framed by the paper.

**Requested Changes:**

1. Can you provide a Pareto analysis across budgets with frontier plots and marginal gain per unit cost, plus a short guide for choosing a budget point.

2. Please run robustness studies on the worker model by simulating correlated annotators and mixed skill groups, then report the stability of allocations and sensitivity to misspecified item difficulty.

3. Is it possible for you to add calibration checks for the difficulty proxy. Compare raw probabilities with temperature calibrated versions, test both weaker and stronger proxy models, and show the impact on the Q table and the final curves.

4. Can you include a small real platform study. Compare uniform allocation, a simple sequential stopping baseline, BUOCA, and the integer program when feasible. Report accuracy, unit cost, and latency distributions.

5. Strengthen statistical reporting with confidence intervals at representative budgets and simple significance tests at selected points. Identify budget regions where marginal gains become small for decision making.

6. Can you offer a brief multi-class theory note. Provide a data dependent bound or a condition under which the greedy allocation matches the optimum, narrowing the gap between the binary and multiclass results.

---

> ### Author Response · Authors · 2025-11-12
> **Round 1 Response to Reviewer 39Dj**
>
> **Re. Weakness 1 and Requested Changes 1**
>
> The choice of budget will be application dependent. The CLR performance as a function of the budget is concave-shaped (provably so, for the binary case). So there will always be diminishing returns on performance improvements with budget increases (please see Figs. 4, 5). But the steepness of the concave-shaped tradeoff curves will differ from application-to-application. In some cases, the rise may be steep and in others, not so much. As a rough practical recommendation, it would be highly beneficial to increase the budget in the regimes where the slope is high. We could discuss this point in more detail in a revised paper draft.
>
> We do not understand the reviewer’s request related to Pareto analysis. Apologies for the trouble, but could the reviewer kindly elaborate what they have in mind? Don’t Figures 4 and 5 already provide “frontier plots” ? We may be misunderstanding the reviewer’s request.
>
> Please also see our response to “Re. Weakness 8 and Requested Changes 5” below.
>
> **Re. Weakness 2 and Requested Changes 2**
>
> Firstly, we note that the assumptions in our theoretical analysis are motivated by the real-world unavailability of fine-grained information about task-difficulties, labeling noise, or the expertise, biases, and correlation of annotators, etc. For example, on real-world crowdsourcing platforms like Amazon Mechanical Turk, worker correlation information is not available for quality optimization.
>
> We think that modeling worker correlation and studying their effects, even via simulations, can be quite tricky. If the reviewer could suggest some specific models that they think are reasonable or are well accepted in this domain, we could attempt to conduct some simulation-based experiments along those lines. But our current position is that correlation analysis, though very interesting and valuable, is more suitable for future research.
>
> **Re. Weakness 3**
>
> We agree, but would like to reiterate that the assumptions in our theoretical analysis are motivated by the real-world unavailability of fine-grained information about task-difficulties, labeling noise, or the expertise, biases, and correlation of annotators, etc.
>
> **Re. Weakness 4**
>
> We agree and can revise the paper draft to mention this point in a discussion of limitations of the present work. Quantifying the suboptimality of BUOCA for the general multiclass setting is an important direction for future research.
>
> **Re. Weakness 5 and Requested Changes 4**
>
> While acknowledging the reviewer’s point, we note that our paper is theory-focused. In order to test our approach on any dataset, the dataset needs to at least contain a sufficiently large number of worker labels  and ground truths. We are not aware of any significantly larger, publicly available datasets with this property. One could potentially create a new large dataset by carefully designing and conducting new real-world crowdsourcing experiments of the kind suggested by the reviewer. This, however, is highly nontrivial, requiring substantial time and effort beyond our current capability, and outside the scope of the present work. We could, however, overlay the uniform-allocation budget point on the CLR-budget curves shown in Figs. 4, 5.
>
> **Re. Weakness 6 and 7, Summary of Claims, and Broader Impact Concerns**
>
> We agree and can include a paragraph discussing these points under limitations of our work. However, please note that we do provide the standard deviation intervals, in Figure 4 and Figure 5.
>
> **Re. Weakness 8 and Requested Changes 5**
>
> We do provide standard deviation intervals, in Figure 4 and Figure 5. We acknowledge that other statistical measures could potentially be computed. We do not quite understand what the reviewer means by quantifying diminishing returns for decision-making.  We think the concave-shaped  saturating CLR versus budget curves in Figure 4 and Figure 5 make it quite clear where the marginal gains become small. Please also see our response to Weakness 1 above.
>
> **Re. Requested Changes 3**
>
> Could the reviewer kindly clarify what they mean by “temperature calibrated versions” ? In our simulation experiments, we use three model instances (trained for different numbers of epochs) as proxies of weaker/stronger models. The comparisons are made in the subsection on Discussion of Results and in Figure 4.
>
> **Re. Requested Changes 6**
>
> Theoretically quantifying the suboptimality of BUOCA for the general multiclass setting would be valuable, but it requires significant effort beyond our capability to properly address it within the timeline of this review cycle. We believe this is more suitable for future research.

---

### Review · Reviewer_U2vB · 2025-11-09

**Summary Of Contributions:**

This paper studies the problem of worker allocation in a crowdsourcing setting. The problem is posed as an optimization problem, where the goal is to maximize the average accuracy of worker labels, while respecting the budget constraint on worker costs. The authors have proposed a BUOCA algorithm (a polynomial-time greedy algorithm) for binary labels and proved its optimality. Subsequently, the authors extend the algorithm for multiclass classification, and show its utility via experiments.

**Additional Comments:**

My main concern is regarding the absence of discussion on recent work, post-2019. This feels like a major omission.

**Audience:**

Yes

**Audience Explanation:**

Crowdsourcing problems have seen a lot of work in past decade or so. The problem setting studied in the paper (albeit too simplistic), might be interesting.

**Claims And Evidence:**

Yes

**Claims Explanation:**

- The authors prove the global optimality of BUOCA algorithm for binary variables in Section 4.2. They also explain why no. of additional workers assigned to each task during the algorithm increases by 2 at each step (Theorem 4.3).
- Results on multi-class extension are in Section 6.

**Requested Changes:**

Major
- Surprisingly, almost all the references in the paper (except 2) are all pre-2019. This is concerning. What about the work that has happened in recent years? If there is recent work, the authors need to comment on how their work compares to it. If there is none (which I doubt), that raises questions about whether the TMLR audience is really interested in this problem. For example, some papers on a quick search include [1,2].
- I'm also puzzled by the assumptions in Section 3.2. The crowdworkers are assumed i.i.d. since the authors only care about finding the number of workers per task. However, each sample is also assumed i.i.d. In that case, shouldn't $p_i$ (the probability $i$-th sample being labeled correctly) be the same across samples? The authors state that "it is the same no matter what the ground truth label of the sample is or which crowdworker labels it." In combination, Assumption 3.0.1-3.0.3 seem too simplistic.
- Also, give the assumptions, Theorem 4.1 is hardly surprising - if all workers are iid, of course MLE estimate is majority/minority voting (depending on $p_i$).

These are comments based on my very limited knowledge of the recent crowdsourcing literature. I would love to be corrected if I am mistaken.

[1] Ding, Xingjian, et al. "Optimizing worker selection in collaborative mobile crowdsourcing." IEEE Internet of Things Journal 11.4 (2023): 7172-7185.
[2] Hikima, Yuya, et al. "An improved approximation algorithm for wage determination and online task allocation in crowd-sourcing." Proceedings of the AAAI Conference on Artificial Intelligence. Vol. 37. No. 4. 2023.

---

> ### Author Response · Authors · 2025-11-12
> **Round 1 Response to Reviewer U2vB**
>
> **Re. “Surprisingly, almost all the reference … include [1,2].”**
>
> We appreciate the reviewer’s concern over related works. Indeed, there exist many research works with the keyword “crowdsourcing” post 2019. However, in the field of computer science, the meaning and the focus of crowdsourcing has shifted over time. In the past, crowdsourcing primarily meant “hiring online workers to annotate or analyze data.” This practice has been the main method to obtain labels for large machine learning datasets. Later on, the word crowdsourcing evolved into a much broader concept which includes almost any online information-gathering practice such as creating articles on Wikipedia, online voting, reCAPTCHA, and so on. While these practices share a similar process as the crowdsourcing for data annotation, the nature of these tasks create task-specific variables, resulting in very different scenarios when researchers analyze and study their procedures. Take the paper “Optimizing worker selection in collaborative mobile crowdsourcing” by Ding et al. suggested by the reviewer for example. The crowdsourcing in their paper specifically means hiring crowdworkers to use mobile devices to gather real-world environment data. In their crowdsourcing task, the data cannot be evaluated using any ground truth and the crowdworkers are allowed to collaborate. In addition, there is no underlying stochastic model for the generation of samples and labels. These aspects make such crowdsourcing works very different from ours.
>
> Furthermore, the works still related to the old meaning of crowdsourcing largely focus on impacts of generative AI on the crowdsourcing procedures, which is not the focus of our paper. The other paper mentioned by the reviewer, "An improved approximation algorithm for wage determination and online task allocation in crowd-sourcing” by Hikima et al., uses the “adaptive worker assignment” method for task allocation which is distinct from our batch method, as pointed out in our related work section. In fact, the works about task allocation cited in that paper are also pre-2019.
>
> Nonetheless, we thank the reviewer for the suggestion to add more recent works. We can revise the related work section by including more recent discussions on crowdsourcing, including the ones that are not directly related to our problem.
>
> **Re. “I’m also puzzled by … seem too simplistic.”**
>
> In retrospect, we think our wording of assumption 3.0.3 has caused this confusion and it can be clarified with more detailed explanation. Each task is composed of two parts: the content (text, image, etc.) and the label. The assumption means that the probability of correct labeling depends *directly* on the content and not on the label.
>
> In Assumption 3.0.1, crowdworkers make iid decisions when annotating each sample. The probability that a decision will be correct depends on the sample. For any given sample, the probability of correctly labeling it will be the same for any crowdworker (statistical indistinguishability of crowdworker decision-making), but the value of that probability will depend on the content of the sample being annotated.
>
> In assumption 3.0.2, by “are sampled in an iid manner according to some underlying distribution,” we mean that all workers receive tasks completely at random (neither the task requester nor the platform will inject any rules in distributing the tasks).

---

### Decision · Action_Editor_PYt8 · 2025-12-30

**Recommendation:** Accept with minor revision

**Additional Comments:**

While I am largely satisfied with the changes made as a result of the discussion, there is one issue I do not think has been fully captured.  Specifically, Weakness 4 from Reviewer 39Dj, that the approximation performance of multi-class BUOCA is purely an empirical claim, should be made explicit in the conclusion.  Modulo that minor addition the paper appears to meet TMLR's standards and needs only be put into the final, de-anonymized format for acceptance.

However, Reviewer 39Dj commented on, and another reviewer expressed interest in, the issue of choosing a budget.  I also find this question interesting.  I agree with the authors that Figure 4 presents an adequate tradeoff curve between budget and performance.  However, (a) the text accompanying the figure only discusses the epochs and number of workers.  It would be nice to have some discussion about budgets here.  (b) Figure 4 suggests there is a ``sweet spot'' before which adding budget has a consistent improvement in performance and after which it has very limited effect.  Is there any natural way to tune BUOCA to be near this point?  While this is not a strictly necessary change to meet TMLR's standards, adding a paragraph here shouldn't be too much work and would empirically be of interest to several readers so I strongly encourage the authors to do so.

**Audience:**

Yes

**Audience Explanation:**

All three reviewers agree that the paper may be of interest to those who study crowdsourcing labeled data and related topics.

**Claims And Evidence:**

Yes

**Claims Explanation:**

Reviewers raised a number of specific concerns in their original reviews which all appear to have been clarified or addressed during the discussion process.  Some reviewers have remaining concerns about the evidence for the external validity of the the results and the applicability to practice.  These are fair criticisms of the work, but as the authors note the paper is theory-focused.  I believe it provides sufficient evidence for the claims it does make in this regard and agree with the authors that the remaining concerns are best viewed as limitations and opportunities for future work.